# Exact Recovery of Sparse Binary Vectors from Generalized Linear Measurements

**Arya Mazumdar** [1]  **Neha Sangwan** [1]

## Abstract

We consider the problem of *exact* recovery of a $k$-sparse binary vector from generalized linear measurements (such as *logistic regression*). We analyze the *linear estimation* algorithm (Plan, Vershynin, Yudovina, 2017), and also show information theoretic lower bounds on the number of required measurements. As a consequence of our results, for noisy one bit quantized linear measurements (1bCSbinary), we obtain a sample complexity of $O((k + \sigma^2) \log n)$, where $\sigma^2$ is the noise variance. This is shown to be optimal due to the information theoretic lower bound. We also obtain tight sample complexity characterization for logistic regression. Since 1bCSbinary is a strictly harder problem than noisy linear measurements (SparseLinearReg) because of added quantization, the same sample complexity is achievable for SparseLinearReg. While this sample complexity can be obtained via the popular lasso algorithm, linear estimation is computationally more efficient. Our lower bound holds for any set of measurements for SparseLinearReg (similar bound was known for Gaussian measurement matrices) and is closely matched by the maximum-likelihood upper bound. For SparseLinearReg, it was conjectured in Gamarnik and Zadik, 2017 that there is a statistical-computational gap and the number of measurements should be at least $(2k + \sigma^2) \log n$ for efficient algorithms to exist. It is worth noting that our results imply that there is no such statistical-computational gap for 1bCSbinary and logistic regression.

---

[1]Halıcıoğlu Data Science Institute, University of California San Diego, La Jolla, United States. Correspondence to: Neha Sangwan <nehasangwan010@gmail.com>.

*Proceedings of the 42nd International Conference on Machine Learning*, Vancouver, Canada. PMLR 267, 2025. Copyright 2025 by the author(s).

## 1. Introduction

Sparse linear regression and compressed sensing have been a topic of intense research in statistics and signal processing for the past few decades (Candès et al., 2006; Donoho, 2006; Tibshirani, 1996; Do Ba et al., 2010). The problem of **binary** sparse linear regression (SparseLinearReg) considers linear measurements of an unknown binary vector, corrupted by additive Gaussian noise. Focusing on binary signals, this particular problem has recently been studied in (Gamarnik & Zadik, 2017a;b; 2022; Reeves et al., 2019), mainly motivated by the question of *support recovery* of sparse signals (Wainwright, 2009). Formally, for an unknown $k$-sparse signal $\mathbf{x} \in \{0,1\}^n$, a sensing matrix $\mathbf{A} \in \mathbb{R}^{m \times n}$ and a noise vector $\mathbf{z} = (z_1, \ldots, z_m)$ where $z_i$s are iid $\mathcal{N}(0, \sigma^2)$ for some variance $\sigma^2$, we observe $\mathbf{y}$ given by

$$\mathbf{y} = \mathbf{A}\mathbf{x} + \mathbf{z}. \tag{1}$$

Our goal is to design the (possibly random) sensing matrix $\mathbf{A}$ with a power constraint, i.e.,

$$\mathbb{E}[(\mathbf{A}_i^T \mathbf{x})^2] \leq k, i = 1, \ldots, m, \tag{2}$$

where $\mathbf{A}_i$ denotes $i^{th}$ row of the matrix $\mathbf{A}$ and the expectation is over the possible randomness in $\mathbf{A}$, and a decoding algorithm $\phi$ such that

$$\max_{\mathbf{x} \in \{0,1\}^n, |\mathbf{x}|_{\mathsf{H}} = k} \mathbb{P}\left(\phi(\mathbf{A}, \mathbf{y}) \neq \mathbf{x}\right) \to 0 \text{ as } n \to \infty. \tag{3}$$

Here, $|\mathbf{x}|_{\mathsf{H}}$ denotes the Hamming weight of $\mathbf{x} \in \{0,1\}^n$. The probability is computed over the randomness of the sensing matrix and the (randomized) algorithm.

The problem of one bit quantized linear measurements (also known as one bit compressed sensing (1bCSbinary)) is similar, except that the output vector $\mathbf{y}$ is the sign of $\mathbf{A}\mathbf{x} + \mathbf{z}$ instead of the entire vector $\mathbf{A}\mathbf{x} + \mathbf{z}$ (Boufounos & Baraniuk, 2008). That is, we observe

$$\mathbf{y} = \mathsf{sign}(\mathbf{A}\mathbf{x} + \mathbf{z}). \tag{4}$$

Here, $\mathbf{y} = (y_1, \ldots, y_m)$ is defined as $y_i = \mathsf{sign}(\mathbf{A}_i^T \mathbf{x} + z_i)$, $i \in [1 : m]$ where $\mathsf{sign}(a) = 1$ if $a \geq 0$ and $\mathsf{sign}(a) = -1$

otherwise. An algorithm $\phi'$ for 1bCSbinary takes input $\mathbf{y}$ and $\mathbf{A}$. Again, we require that [1]

$$\max_{\mathbf{x} \in \{0,1\}^n, |\mathbf{x}|_{\mathsf{H}}=k} \mathbb{P}\left(\phi'(\mathbf{A}, \mathbf{y}) \neq \mathbf{x}\right) \to 0 \text{ as } n \to \infty. \quad (5)$$

Usually in 1-bit compressed sensing, the Gaussian noise before quantization is not present. Our formulation can be considered as a *sparse* "probit model" (McCullagh, 2019).

More generally, we define the problem of generalized linear measurements (GLMs) , e.g., (Kakade et al., 2011; Plan et al., 2017) where we assume that the observation $\mathbf{y} = (y_1, \dots, y_m)$ is related to the sparse binary input vector $\mathbf{x}$ using an "inverse link" function $g$ such that for each $i \in [m]$,

$$\mathbb{E}\left[y_i | \mathbf{A}_i\right] = g\left(\mathbf{A}_i^T \mathbf{x}\right). \quad (6)$$

That is, the expected value of the output $y_i$ is linked to $\mathbf{A}_i$ only through $\mathbf{A}_i^T \mathbf{x}$. For example, for SparseLinearReg

$$\mathbb{E}\left[y_i | \mathbf{A}_i\right] = \mathbf{A}_i^T \mathbf{x},$$

for 1bCSbinary

$$\mathbb{E}\left[y_i | \mathbf{A}_i\right] = 1 - 2\Phi\left(\frac{-A_i^T \mathbf{x}}{\sigma}\right)$$

where $\Phi$ is the Gaussian cumulative distribution function.

In the logistic regression model (LogisticRegression), we observe a binary output $y_i \in \{-1, 1\}$ for each measurement $i \in [m]$. The probability that $y_i$ takes value 1 is given by

$$\mathbb{P}\left(y = 1\right) = \frac{1}{1 + e^{-\beta \mathbf{a}^T \mathbf{x}}}.$$

for parameter $\beta > 0$. The parameter $\beta$ controls the level of noise. When $\beta \to \infty$, the model approaches noiseless one bit compressed sensing. As $\beta$ decreases, the output becomes more noisy. When $\beta = 0$, the output is uniformly distributed on $\{-1, 1\}$ and is independent of $\mathbf{x}$. In this model,

$$\mathbb{E}\left[y_i | \mathbf{A}_i\right] = \tanh\frac{\beta \mathbf{A}_i^T \mathbf{x}}{2}.$$

**Our contributions.** In this paper, our contributions are the following:

- We analyze the linear estimation+projection algorithm (Plan et al., 2017) for generalized linear measurements of sparse binary inputs (Theorem 2.1). We also provide an information theoretic lower bound (Theorem 2.5).

- As corollaries, we obtain tight sample complexity characterization for noisy one bit compressed sensing (Corollary 2.2 and Corollary 2.6) and logistic regression (Corollary 2.4 and Corollary 2.7).

- The algorithm can be used for SparseLinearReg either directly (Corollary 2.3) or by first quantizing the received signal to its sign value and then using the algorithm for 1bCSbinary. The sample complexity is the same for both these cases. This shows that in the regime where the number of measurements are at least $C(k + \sigma^2) \log n$ for some constant $C$, keeping only the sign information is sufficient for SparseLinearReg.

- We provide "almost" matching information theoretic lower (Corollary 2.8) and upper bounds (Theorem 2.9) for exact recovery in SparseLinearReg. If the measurements are Gaussian, we get slightly better lower bounds (Theorem 2.10).

A summary of results in presented in Tables 1, 2 and 3.

## 1.1. Discussion of results and related works

**Intuitions on lower bounds.** Observe that 1bCSbinary is a strictly more difficult problem than SparseLinearReg in the sense that any algorithm that works for 1bCSbinary can be used for SparseLinearReg by using only the sign information. Thus, the sample complexity of 1bCSbinary is at least as much as SparseLinearReg, the latter can be much smaller in some cases. From an information theoretic viewpoint, a randomly chosen $k$-sparse vector $\mathbf{x}$ has entropy $\log \binom{n}{k} \approx k \log \frac{n}{k}$. Since each $y_i$ can give at most one bit of information, we need at least $k \log \frac{n}{k}$ measurements for 1bCSbinary (See Corollary 2.6 for the exact lower bound) to learn $\mathbf{x}$. For SparseLinearReg on the other hand, the output has infinite precision. In fact, we can show that in the absence of noise, only one sample is sufficient to recover the unknown signal (see Remark 1.1). SparseLinearReg can be viewed as a coding problem for a Gaussian channel, where $\mathbf{x}$ is the message and $\mathbf{A}\mathbf{x}$ is its corresponding codeword. Thus, from the converse for Gaussian channel (see (Cover & Thomas, 2006, Theorem 9.1.1)), we need at least $\frac{k \log(n/k)}{C}$ samples for exact recovery. Here $C$ is the capacity of the Gaussian channel, which depends on SNR (a function of $\mathbf{A}\mathbf{x}$ and $\sigma^2$). Given the power constraint of Eq. (2) (which is satisfied when entries of $\mathbf{A}$ are chosen iid $\mathcal{N}(0, 1)$), the capacity $C$ is $\frac{1}{2} \log\left(1 + \frac{k}{\sigma^2}\right)$, thereby showing that the lower bound for SparseLinearReg can be much smaller.

**Binary sparse linear regression.** The problem of binary sparse linear regression was introduced in (Gamarnik & Zadik, 2017a; 2022) and was further studied in (Reeves et al., 2019). An *"all or nothing"* phenomenon was shown in (Reeves et al., 2019) for *approximate recovery* of binary vec-

---

[1]The probability of error measured by (5) corresponds to the *'for each'* criterion in the one bit compressed sending literature. The *'for all'* criterion which requires that the same sensing matrix works for all unknown signals corresponds to showing $\mathbb{P}\left(\exists \mathbf{x} \text{ such that } \phi'(\mathbf{A}, \mathbf{y}) \neq \mathbf{x}\right) \to 0 \text{ as } n \to \infty.$

tors at the critical sample complexity of $m^* \triangleq \frac{2k \log n/k}{\log\left(1 + \frac{k}{\sigma^2}\right)}$, showing that approximate recovery is possible if and only if $m \geq m^*$. It was additionally conjectured in (Gamarnik & Zadik, 2017a) that no efficient algorithms exist in the regime $m^* \leq m \leq m_{\text{alg}} \triangleq (2k + \sigma^2) \log n$. When $m \geq m_{\text{alg}}$, various algorithms like Lasso (Wainwright, 2009), Orthogonal Matching Pursuit (OMP) (Tropp & Gilbert, 2007) and (Ndaoud & Tsybakov, 2020) can recover the sparse vector. It has also been shown in (Gamarnik & Zadik, 2017b) that lasso fails to recover unknown vector $\mathbf{x}$ when $m \leq c\, m_{\text{alg}}$ for some small constant $c$. Outside this regime, a local search algorithm was proposed (Gamarnik & Zadik, 2017b), which starts with a guess of $\mathbf{x}$ and iteratively updates it.

In (Reeves et al., 2019), the information theoretic lower bound of $m^*$ is shown for the case when each entry of the sensing matrix is chosen iid $\mathcal{N}(0, 1)$. We consider the exact recovery guarantee for the problem and show that $m \geq m^*$ samples are necessary even when the sensing matrix is not Gaussian (Corollary 2.8)[2]. We show an almost matching upper bound based on the Maximum Likelihood Estimator (MLE) using a random Gaussian sensing matrix (Theorem 2.9 and Theorem 2.10). This is along the lines of the MLE analysis in (Reeves et al., 2019), which was done for approximate recovery (our sample complexity for exact recovery turns out to be slightly different).

*Remark* 1.1. It was observed in (Gamarnik & Zadik, 2017a) that in the no-noise regime ($\sigma^2 = 0$), one measurement is sufficient to recover the underlying vector by brute force. However, it is conjectured that there is no efficient algorithm if $m \leq 2k \log n$. The results in (Gamarnik & Zadik, 2017a) were shown only when the entries of the sensing matrix are chosen iid $\mathcal{N}(0, 1)$ (i.e. Gaussian design). For an arbitrary sensing matrix, an efficient way to recover $\mathbf{x}$ using only one measurement is by using $\mathbf{A} = \frac{1}{2^n}[1, 2, 2^2, \ldots, 2^{n-1}]$. Note that $2^n \times \mathbf{y}$ in this case is the value of unknown signal in the decimal system (base 10). It can be converted to binary in $O(n)$ time. This suggests that for specific non-random constructions, there may be efficient algorithms in the conjectured hardness regime.

**Binary one-bit compressed sensing.** The problem of one bit compressed sensing has been well studied e.g. (Boufounos & Baraniuk, 2008; Jacques et al., 2013) including greedy algorithms (e.g. (Liu et al., 2016)) and noisy test outcomes (e.g. (Matsumoto & Mazumdar, 2024)), and the problem of recovering binary vectors has also been studied in (Acharya et al., 2017; Mazumdar & Pal, 2022). However, these works do not consider the Gaussian noise prior to quantization. The best known upper bound ($O(k/\epsilon)$

from (Matsumoto & Mazumdar, 2022)) when specialized to exact recovery for binary sparse vectors requires $O(k^{3/2})$ (by choosing $\epsilon = 1/\sqrt{k}$). On the other hand, our bound is $O(k \log n)$. This discrepancy is because the previous models are studied for the "for all" model which is a harder problem than our present "for each" model. The results in (Plan et al., 2017), on the other hand, are for the "for each" model, though their analysis is not optimal for binary vectors (see Appendix B). The problem of noisy one bit compressed sensing (1bCSbinary) introduced here is motivated by the probit model (e.g. see a modern treatments of the non-sparse probit model (Kuchelmeister & van de Geer, 2024)). Here we provide an information theoretic lower bound of $m \geq (k + \sigma^2) \log(n/k)$ and show that the aforementioned efficient algorithm (Algorithm 1) works with the same $m = O((k + \sigma^2) \log n)$ samples and has a computational complexity of $O((k + \sigma^2)n \log n)$. We also provide optimal sample complexity characterization for learning binary sparse vectors under the logistic regression model, which was previously studied for learning real vectors (Hsu & Mazumdar, 2024; Plan et al., 2017).

**Algorithm for binary vectors.** We consider a simple algorithm which is equivalent to the "average algorithm" (Servedio, 1999) or "linear estimator" (Plan et al., 2017), followed by a selection of the 'top-k' coordinates. Regarding the intuition behind the algorithm, we observe that for an unknown signal $\mathbf{x}$, the output $\mathbf{y}$ and $\mathbf{A}_{\mathcal{S}_{\mathbf{x}}}$, the restriction of the sensing matrix to columns where $\mathbf{x}$ is 1, are correlated whereas $\mathbf{y}$ and $\mathbf{A}_{[1:n] \setminus \mathcal{S}_{\mathbf{x}}}$ are uncorrelated. Here, $\mathbf{A}_{[1:n] \setminus \mathcal{S}_{\mathbf{x}}}$ denotes the restriction of the sensing matrix to columns where $\mathbf{x}$ is 0. Thus, we compute the inner product between $\mathbf{y}$ and each column of the sensing matrix as a proxy for correlation between the output and the corresponding column. The output of the algorithm is the top $k$-most correlated columns (See Algorithm 1 for details.). One can also think of this as a "one-shot" version of the popular OMP algorithm. This algorithm requires $O((k + \sigma^2) \log n)$ samples for 1bCSbinary and SparseLinearReg and $O((k+1/\beta^2) \log n)$ for LogisticRegression. It has a computation complexity of $O((k + \sigma^2)n \log n)$. Most of the previous algorithms, including the one in (Plan et al., 2017), were given for the case when the unknown signal is not necessarily binary. It should be noted that the black-box application of the result of (Plan et al., 2017) specialized to binary inputs will not recover the optimal sample complexity. See Appendix B where we show that the results in (Plan et al., 2017) imply a sample complexity of $O(k^2 \log(2n/k))$. We provide a simple yet optimal analysis of the sample complexity in our special case of sparse binary signals.

We would like to emphasize that the sample complexity of Algorithm 1 for both SparseLinearReg and 1bCSbinary is the same ($O((k + \sigma^2) \log n)$). This implies that for

---

[2] Our lower bounds hold for a weaker average probability of error recovery criteria, instead of the maximum probability of error in Eq. (3). However, for random Gaussian sensing matrix both criterion can be shown to be equivalent.

SparseLinearReg, when $m$ is outside the conjectured hardness regime, we do not need the amplitude of $\mathbf{y}$, only the sign information is sufficient to recover the unknown signal.

| | |
|---|---|
| Upper bound (efficient algorithms) (exact recovery) | $O\left(\left(k+\sigma^2\right)\log\left(n\right)\right)$ (Lasso, OMP and this paper) |
| Upper bound (MLE) (exact recovery) | $\max_{l\in[1:k]}\frac{nN(l)}{\frac{1}{2}\log\left(\frac{l}{2\sigma^2}+1\right)}$ (this paper ) |
| Lower bound (exact recovery) | $\max_{l\in[1:k]}\frac{nN(l)}{\frac{1}{2}\log\left(1+\frac{l}{\sigma^2}\left(2-\frac{l}{k}\right)\right)}$ (this paper ) |
| Upper bound (MLE) (approximate recovery) | $\frac{2k\log n/k}{\log\left(1+\frac{k}{\sigma^2}\right)}$ ((Reeves et al., 2019) ) |
| Lower bound (approximate recovery) | $\frac{2k\log n/k}{\log\left(1+\frac{k}{\sigma^2}\right)}$ ((Reeves et al., 2019) ) |

*Table 1.* The table gives the upper and lower bounds on the sample complexity of sparse linear regression. Note that $N(l) := \frac{k}{n}h_2\left(\frac{l}{k}\right) + (1-\frac{k}{n})h_2\left(\frac{l}{n-k}\right)$ where $h_2(\cdot)$ is the binary entropy function.

| | |
|---|---|
| Upper bound (efficient algorithm) | $O\left(\left(k+\sigma^2\right)\log\left(n\right)\right)$ (this paper) |
| Lower bound | $\frac{k+\sigma^2}{2}\log\left(\frac{n}{k}\right)$ (this paper ) |

*Table 2.* The table gives the upper and lower bounds on the sample complexity of one bit compressed sensing for exact recovery.

| | |
|---|---|
| Upper bound (efficient algorithm) | $O\left(\left(k+\frac{1}{\beta^2}\right)\log\left(n\right)\right)$ (this paper) |
| Lower bound | $\frac{1}{2}\left(k+\frac{1}{\beta^2}\right)\log\left(\frac{n}{k}\right)$ (this paper ) |

*Table 3.* The table gives the upper and lower bounds on the sample complexity of logistic regression for exact recovery.

**Notation.** We denote the set of integers $\{1,2,\ldots,n\}$ interchangeably by $[n]$ and $[1:n]$. We will use boldfaced uppercase letters like $\mathbf{A}$ for matrices and lowercase letters such as $\mathbf{x}$ for vectors. The entry of the matrix at $i^{\text{th}}$ row and $j^{\text{th}}$ column is denoted by $A_{i,j}$. Similarly, the $i^{\text{th}}$ entry of a vector $\mathbf{x}$ is denotes by $x_i$. For any binary vector $\mathbf{x} = (x_1,\ldots,x_n)$, we denote the set of indices $i$ where $x_i = 1$ by $\mathcal{S}_{\mathbf{x}} \subseteq [1:n]$ and we use $\mathbf{A}_{\mathcal{S}_{\mathbf{x}}}$ to denote the restriction of $\mathbf{A}$ to the columns where $\mathbf{x}$ is 1. We use $\mathbf{A}_i$ to denote $i^{\text{th}}$ row and $\mathbf{A}^{(i)}$ to denote $i^{\text{th}}$ column. We denote the binary entropy function by $h_2(\cdot)$ and differential entropy

by $h(\cdot)$. The subgaussian norm of a random variable $w$ is denoted by $\|w\|_{\psi_2}$.

**Organization.** We present the algorithm and upper bounds in Section 2.1. The information theoretic lower bounds are presented in Section 2.2. In Section 2.3, we present an upper bound for SparseLinearReg based on the maximum likelihood estimator. We also provide a lower bound in this section, which closely matches the upper bound. This lower bounds does not follow as a corollary to the general lower bound theorem for GLMs (Theorem 2.5). It requires a separate analysis based on a conditional version of Fano's inequality. We provide proofs of the upper and lower bound for GLMs (Theorems 2.1 and 2.5) in Section 3. Remaining proofs are delegated to Appendix A. We provide detailed comparison of our results with (Plan et al., 2017) in Appendix B. We conclude with a discussion on open problems in Section 4.

## 2. Main results

### 2.1. Algorithm

We analyze the simple linear estimation based algorithm from (Plan et al., 2017) for generalized linear measurements, specializing it for binary vectors. The algorithm (Algorithm 1) takes the sensing matrix $\mathbf{A}$ and the output vector $\mathbf{y}$ as the inputs. For each column $\mathbf{A}^{(i)}$, $i \in [1:n]$ of the sensing matrix, the algorithm computes $l_i = \langle \mathbf{y}, \mathbf{A}^{(i)} \rangle = \sum_{j=1}^m y_j A_{j,i}$ where $A_{j,i}$ is the entry at $j^{\text{th}}$ row and $i^{\text{th}}$ column.

The vector $\mathbf{l} = (l_1,\ldots,l_n)$ is then sorted in decreasing order. The output of the algorithm is a set containing the indices of the top-$k$ elements of the sorted vector. That is, if the sorted vector is $(l_{\alpha_1}, l_{\alpha_2}, \ldots, l_{\alpha_n})$ where $l_{\alpha_i} \geq l_{\alpha_j}$ for $i \leq j$, then the output of the algorithm is $\mathcal{S} = \{\alpha_1, \ldots, \alpha_k\}$.

---

**Algorithm 1** Top-$k$ correlated indices

**Input:** Sensing matrix $\mathbf{A} \in \mathbb{R}^{m\times n}$ and output $\mathbf{y} \in \mathbb{R}^m$
**Output:** a $k$-sized subset of $[1:n]$
$\mathbf{l} \leftarrow (0,\ldots,0)$, $\mathbf{l} \in \mathbb{R}^n$
**for** each $i \in [1:n]$ **do**
$\quad l_i \leftarrow \sum_{j=1}^m y_j A_{j,i}$
**end for**
Sort $\mathbf{l}$ in decreasing order and let $\mathcal{S}$ be the top $k$ indices.
**Return:** $\mathcal{S}$

---

The convergence and sample complexity guarantees for the algorithm are shown for the case when each entry of $\mathbf{A}$ is chosen iid $\mathcal{N}(0,1)$. Note that such a matrix satisfies the power constraint in (2). As we argued in Section 1, for the unknown signal $\mathbf{x}$, the output $\mathbf{y} = \mathbf{A}\mathbf{x} + \mathbf{z}$ is correlated with each column $\mathbf{A}^{(i)}$ for $i \in \mathcal{S}_{\mathbf{x}}$ and uncorrelated with $\mathbf{A}^{(j)}$ for $j \notin \mathcal{S}_{\mathbf{x}}$. In particular, for large number of samples, when $i \in$

$\mathcal{S}_\mathbf{x}$, the inner product $\langle \mathbf{y}, \mathbf{A}^{(i)} \rangle$ is close to $\mathbb{E}\left[\langle \mathbf{y}, \mathbf{A}^{(i)} \rangle\right] = m$ (for linear regression) with high probability. On the other hand, $\langle \mathbf{y}, \mathbf{A}^{(j)} \rangle$ is close to 0 for $j \notin \mathcal{S}_\mathbf{x}$. Thus, $l_i$ for $i \in \mathcal{S}_\mathbf{x}$ will dominate over $l_j$ for $j \notin \mathcal{S}_\mathbf{x}$. This line of argument also works when the output is binary, though in this case $\mathbb{E}\left[\langle \mathbf{y}, \mathbf{A}^{(i)} \rangle\right]$ for $i \in \mathcal{S}_\mathbf{x}$ is different. This is the main idea of Algorithm 1. We first present Theorem 2.1 for generalized linear measurements.

**Theorem 2.1** (Sample Complexity of Algorithm 1 for GLMs). *Suppose the GLM is such that for each $i \in [m]$, $y_i$ is a subgaussian random variable with subgaussian norm given by $\|y_i\|_{\psi_2}$. For any $\mathbf{x}$, suppose for some $L$, $\mathbb{E}\left[g'(\mathbf{A}_i^T \mathbf{x})\right] \geq L \cdot \|y_i\|_{\psi_2}$ for all $i \in [m]$. Algorithm 1 recovers the unknown signal with high probability if*

$$m \geq \frac{C}{\min\{L, L^2\}}(\log(k) + \log(n - k)) \qquad (7)$$

*where $C$ is some constant.*

When $y_j$ is subgaussian, $y_j \mathbf{A}_{i,j}$ for any $i, j$ is a sub-exponential random variable. This observation allows us to use a concentration result for sub-exponential random variables to analyse the sample complexity. See Section 3 for a detailed proof[3].

As corollaries to Theorem 2.1, we obtain the following sample complexity bounds for 1bCSbinary and SparseLinearReg. These corollaries are proved in Appendix A.1.

**Corollary 2.2** (Sample Complexity of Algorithm 1 for 1bCSbinary). *Algorithm 1 recovers the unknown signal for 1bCSbinary with high probability if $m \geq c_1\left(k + \sigma^2\right)(\log(k) + \log(n - k))$ for some constant $c_1$.*

**Corollary 2.3** (Sample Complexity of Algorithm 1 for SparseLinearReg). *Algorithm 1 recovers the unknown signal for SparseLinearReg if $m \geq c_2\left(k + \sigma^2\right)(\log(k) + \log(n - k))$ for some constant $c_2$.*

Interestingly, the sample complexity for both 1bCSbinary and SparseLinearReg is the same. This can be explained by similar values of $L$, which result in similar rates of concentration of $l_i$'s around their expectation in both the cases. This also implies that in the regime where $m = O((k+\sigma^2)\log(n - k))$, having access to $\mathbf{A}_i^T \mathbf{x} + z_i$ instead of $\text{sign}(\mathbf{A}_i^T \mathbf{x} + z_i)$, does not improve the sample complexity beyond constants.

Using Theorem 2.1, we obtain the following corollary for logistic regression (see proof in Appendix A.1).

---

[3]The current form of Theorem 2.1 is stated for a sensing matrix $\mathbf{A}$ where each entry is chosen iid $\mathcal{N}(0, 1)$. However, the proof technique can be easily generalized for other distributions.

**Corollary 2.4** (Sample Complexity of Algorithm 1 for LogisticRegression). *Algorithm 1 recovers the unknown signal for LogisticRegression if $m \geq c_3\left(k + 1/\beta^2\right)(\log k + \log(n - k))$ for some constant $c_3$.*

Comparing the sample complexity bounds of 1bCSbinary and LogisticRegression, we notice that the sample complexity is similar except that the noise variance $\sigma^2$ is replaced by $1/\beta^2$. This relationship is not surprising as a similar relationship was also present in the sample complexity bounds in (Hsu & Mazumdar, 2024) (for logistic regression) and (Kuchelmeister & van de Geer, 2024) (for probit model). Note that, in the noiseless case, when $\beta \to \infty$ (or $\sigma = 0$ for 1bCSbinary), the sample complexity is $O(k \log n)$, which is close to the simple counting lower bound of $k \log n/k$. On the other hand, when $\beta = 0$ (or $\sigma \to \infty$ for 1bCSbinary), $m \to \infty$, which makes intuitive sense as very high levels of noise render the output useless.

To compute the time complexity of the algorithm, notice that the for loop in step 2 takes $O(n \times m)$ time and step 4 takes $O(n \log n)$ time. Thus, the computational complexity of the algorithm is $O(nm + n \log n)$, which is $O((k + \sigma^2)n \log n)$ for $m = O((k + \sigma^2)) \log n$. To compute the time complexity of the algorithm, notice that the for loop in step 2 takes $O(n \times m)$ time and step 4 takes $O(n \log n)$ time. Thus, the computational complexity of the algorithm is $O(nm + n \log n)$, which is $O((k + \sigma^2)n \log n)$ for $m = O((k + \sigma^2)) \log n$.

### 2.2. Lower bounds on sample complexity

We establish a lower bound for generalized linear measurements using standard information-theoretic arguments based on Fano's inequality. While the upper bound in Theorem 2.1 is derived for the maximum probability of error over all $k$-sparse vectors, the lower bound applies even in the weaker setting of the average probability of error, where $\mathbf{x}$ is chosen uniformly at random.

**Theorem 2.5** (Lower bound for GLMs). *Consider any sensing matrix $\mathbf{A}$. For a uniformly chosen $k$-sparse vector $\mathbf{x}$, an algorithm $\phi$ satisfies*

$$\mathbb{P}\left(\phi(\mathbf{A}, \mathbf{y}) \neq \mathbf{x}\right) \leq \delta$$

*only if the number of measurements*

$$m \geq \frac{k \log\left(\frac{n}{k}\right)}{I}\left(1 - \frac{h_2(\delta) + \delta k \log n}{k \log n/k}\right)$$

*for some $I$ such that $I \geq I(y_i; \mathbf{x}|\mathbf{A})$, $i \in [m]$. In particular, when $y \in \{-1, 1\}$, we have $\mathbb{E}\left[\left(g(\mathbf{A}_i^T \mathbf{x})\right)^2\right] \geq I(y_i, \mathbf{x}|\mathbf{A})$ where the expectation is over the randomness of $\mathbf{A}$ and $\mathbf{x}$.*

The lower bound can be interpreted in terms of a communication problem, where the input message $\mathbf{x}$ is encoded to

$\mathbf{A}\mathbf{x}$. The decoding function takes in as input the encoding map $\mathbf{A}$ and the output vector $\mathbf{y}$ in order to recover $\mathbf{x}$ with high probability. For optimal recovery, one needs at least $\frac{\text{message entropy}}{\text{capacity}}$ number of measurements (follows from noisy channel coding theorem (Cover & Thomas, 2006)). In Theorem 2.5, the entropy of the message set $\log \binom{n}{k} \approx k \log n/k$ and the proxy for capacity is the upper bound on mutual information $I$. We provide a detailed proof of the theorem in Section 3.

We first present lower bounds for 1bCSbinary and LogisticRegression. The lower bound for 1bCSbinary is given for any sensing matrix $\mathbf{A}$ which satisfies the power constraint given by (2), whereas the one for LogisticRegression is only for the special case when each entry of the sensing matrix is iid $\mathcal{N}(0,1)$. Recall that (2) holds in this case. For 1bCSbinary (and LogisticRegression respectively), we can use the upper bound of $\mathbb{E}\left[\left(g(\mathbf{A}_i^T \mathbf{x})\right)^2\right]$ on the mutual information term. The dependence of $\sigma^2$ (and $1/\beta^2$ respectively) requires careful bounding of this term, which is done in the formal proofs in Appendix A.2.

As mentioned earlier, we need at least $k \log(n/k)$ measurements for 1bCSbinary and LogisticRegression. This is because the entropy of a randomly chosen $k$-sparse vector is approximately $k \log(n/k)$ and we learn at most one bit with each measurement. However, due to corruption with noise, we learn less than a bit of information about the unknown signal with each measurement. The information gain gets worse as the noise level increases. Our lower bounds make this reasoning explicit.

**Corollary 2.6** (1bCSbinary lower bound). *Suppose, each row $\mathbf{A}_i$, $i \in [1 : m]$ of the sensing matrix $\mathbf{A}$ satisfies the power constraint (2). For a uniformly chosen $k$-sparse vector $\mathbf{x}$, an algorithm $\phi$ satisfies*

$$\mathbb{P}\left(\phi(\mathbf{A}, \mathbf{y}) \neq \mathbf{x}\right) \leq \delta$$

*for the problem of* 1bCSbinary *only if the number of measurements*

$$m \geq \frac{k + \sigma^2}{2} \log\left(\frac{n}{k}\right)\left(1 - \frac{h_2(\delta) + \delta k \log n}{k \log n/k}\right).$$

**Corollary 2.7** (LogisticRegression lower bound). *Consider a Gaussian sensing matrix $\mathbf{A}$ where each entry is chosen iid $N(0,1)$. For a uniformly chosen $k$-sparse vector $\mathbf{x}$, an algorithm $\phi$ satisfies*

$$\mathbb{P}\left(\phi(\mathbf{A}, \mathbf{w}) \neq \mathbf{x}\right) \leq \delta$$

*for the problem of* LogisticRegression *only if the number of measurements*

$$m \geq \frac{1}{2}\left(k + \frac{1}{\beta^2}\right) \log\left(\frac{n}{k}\right)\left(1 - \frac{h_2(\delta) + \delta k \log n}{k \log n/k}\right).$$

Theorem 2.5 also implies an information theoretic lower bound for SparseLinearReg, which is presented below and proved in Appendix A.2. Note that the denominator term in the bound $\frac{1}{2} \log\left(1 + \frac{k}{\sigma^2}\right)$ is the capacity of a Gaussian channel with power constraint $k$ and noise variance $\sigma^2$.

**Corollary 2.8** (SparseLinearReg lower bound). *Under the average power constraint* (2) *on $\mathbf{A}$, for a uniformly chosen $k$-sparse vector $\mathbf{x}$, an algorithm $\phi$ satisfies*

$$\mathbb{P}\left(\phi(\mathbf{A}, \mathbf{y}) \neq \mathbf{x}\right) \leq \delta$$

*only if the number of measurements*

$$m \geq \frac{k \log\left(\frac{n}{k}\right) - (h_2(\delta) + \delta k \log n)}{\frac{1}{2} \log\left(1 + \frac{k}{\sigma^2}\right)}.$$

### 2.3. Tighter upper and lower bounds for SparseLinearReg

We present information theoretic upper and lower bounds for SparseLinearReg in this section. Similar to Section 2.1, our upper bound is for the maximum probability of error, while the lower bounds hold even for the weaker criterion of average probability of error.

We first present an upper bound based on the maximum likelihood estimator (MLE) where we decode to $\hat{\mathbf{x}}$ if, on output $\mathbf{y}$,

$$\hat{\mathbf{x}} = \arg\max_{\substack{\mathbf{x} \in \{0,1\}^n \\ |\mathbf{x}|_{\mathsf{H}} = k}} p(\mathbf{y}|\mathbf{x})$$

where $p(\mathbf{y}|\mathbf{x})$ denotes the probability density function of $\mathbf{y}$ on input $\mathbf{x}$.

**Theorem 2.9** (MLE upper bound for SparseLinearReg). *Suppose entries of the measurement matrix $\mathbf{A}$ are i.i.d. $\mathcal{N}(0,1)$. The MLE is correct with high probability if*

$$m \geq \max_{l \in [1:k]} \frac{nN(l)}{\frac{1}{2} \log\left(\frac{l}{2\sigma^2} + 1\right)} \tag{8}$$

*where $N(l) := \frac{k}{n} h_2\left(\frac{l}{k}\right) + (1 - \frac{k}{n})h_2\left(\frac{l}{n-k}\right)$.*

We prove the theorem in Appendix A.3. The main proof idea involves analysing the probability that the output of the MLE is $2l$ Hamming distance away from the unknown signal $\mathbf{x}$ for different values of $l \in [1 : k]$ (assuming $k \leq n/2$). This depends on the number of such vectors (approximately $2^{nN(l)}$) and the probability that the MLE outputs a vector which is $2l$ Hamming distance away from $\mathbf{x}$.

Note that when $l = k\left(1 - \frac{k}{n}\right)$, $nN(l) = nh_2(k/n) \approx k \log \frac{n}{k}$ and $\log\left(\frac{k(1-k/n)}{2\sigma^2} + 1\right) \leq \log\left(\frac{k}{2\sigma^2} + 1\right)$. Thus, $m$ is at least $\frac{2k \log n/k}{\log\left(\frac{k}{2\sigma^2} + 1\right)}$ (see the bound for Corollary 2.8).

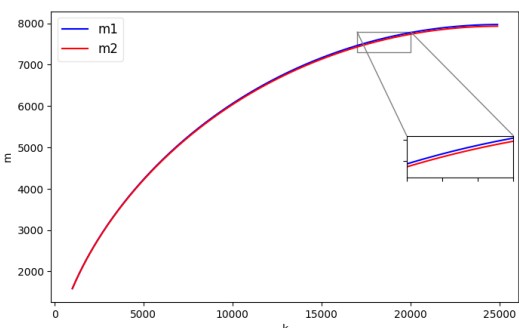

*Figure 1.* The figure shows the plot of the MLE upper bound (8) (given by m1) for different values of $k$. This is displayed in blue color. A plot of $\frac{2nN(l)}{\log\left(\frac{l}{2\sigma^2}+1\right)}$ is also presented for $l = k\left(1 - \frac{k}{n}\right)$ in orange color, given by m2. A part of the plot is zoomed in to emphasize the closeness between the lines. In these plots, $\sigma^2$ is set to 1, $n$ is 50000 and $k$ ranges from 1000 to 25000 ($n/2$).

It is not immediately clear if this value of $l = k\left(1 - \frac{k}{n}\right)$ is the optimizer. However, for large $n$, this appears to be the case numerically as shown in Plot 1.

Inspired by the MLE analysis, we derive a lower bound with the same structure as (8). We generate the unknown signal $\mathbf{x}$ using the following distribution: A vector $\tilde{\mathbf{x}}$ is chosen uniformly at random from the set of all $k$-sparse vectors. Given $\tilde{\mathbf{x}}$, the unknown input signal $\mathbf{x}$ is chosen uniformly from the set of all $k$-sparse vector which are at a Hamming distance $2l$ from $\mathbf{x}$. The lower bound is then obtained by computing upper and lower bounds on $I(\mathbf{A}, \mathbf{y}; \mathbf{x}|\tilde{\mathbf{x}})$. We show this lower bound only for random matrices where each entry is chosen iid $\mathcal{N}(0, 1)$.

**Theorem 2.10** (SparseLinearReg lower bound). *If each entry of $\mathbf{A}$ is chosen iid $\mathcal{N}(0, 1)$, then for a uniformly chosen $k$-sparse vector $\mathbf{x}$, an algorithm $\phi$ satisfies*

$$\mathbb{P}\left(\phi(\mathbf{A}, \mathbf{y}) \neq \mathbf{x}\right) \leq \delta \qquad (9)$$

*only if the number of measurements*

$$m \geq \max_{l \in [1:k]} \frac{nN(l) - 2\log n - h_2(\delta) - \delta k \log n}{\frac{1}{2}\log\left(1 + \frac{l}{\sigma^2}\left(2 - \frac{l}{k}\right)\right)}.$$

The proof of Theorem 2.10 is given in Appendix A.3.

If we choose $l = k\left(1 - \frac{k}{n}\right)$ in Theorem 2.10, we recover corollary 2.8 for the special case of Gaussian design.

## 3. Proofs

*Proof of Theorem 2.1.* Consider any input $\mathbf{x}$ and a sensing matrix $\mathbf{A}$ where each entry is chosen iid $\mathcal{N}(0, 1)$. Suppose $\mathbf{x}$ is supported on $\mathcal{S} \subseteq [1 : n]$ where $|\mathcal{S}| = k$. Let $\mathbf{y} = $

$(y_1, \ldots, y_m)$. Consider the event

$$\mathcal{F} = \left\{\sum_{i=1}^{m} y_i A_{i,j} > \sum_{i=1}^{m} y_i A_{i,j'} \text{ for all } j \in \mathcal{S}, j' \in \mathcal{S}^c\right\}$$

It is clear that under $\mathcal{F}$, the algorithm is correct. We will compute the probability of $\mathcal{F}^c$.

$$
\begin{aligned}
\mathbb{P}\left(\mathcal{F}^c\right) &= \mathbb{P}\left(\bigcup_{j \in \mathcal{S}} \bigcup_{j' \in \mathcal{S}^c} \left\{\sum_{i=1}^{m} y_i A_{i,j'} \geq \sum_{i=1}^{m} y_i A_{i,j}\right\}\right) \\
&\leq \sum_{j \in \mathcal{S}} \sum_{j' \in \mathcal{S}^c} \mathbb{P}\left(\sum_{i=1}^{m} y_i A_{i,j'} \geq \sum_{i=1}^{m} y_i A_{i,j}\right) \\
&= \sum_{j \in \mathcal{S}} \sum_{j' \in \mathcal{S}^c} \mathbb{P}\left(\sum_{i=1}^{m} (y_i(A_{i,j'} - A_{i,j})) \geq 0\right)
\end{aligned}
$$
$$(10)$$

For any $i \in [1 : m]$, $j \in \mathcal{S}$ and $j' \in \mathcal{S}^c$, we first compute $\mathbb{E}\left[y_i(A_{i,j} - A_{i,j'})\right]$.

$$
\begin{aligned}
\mathbb{E}\left[y_i(A_{i,j} - A_{i,j'})\right] &= \mathbb{E}\left[y_i A_{i,j}\right] - \mathbb{E}\left[y_i A_{i,j'}\right] \\
&\overset{(a)}{=} \mathbb{E}\left[y_i A_{i,j}\right] \qquad (11) \\
&\overset{(b)}{=} \frac{\mathbb{E}\left[y_i A_{i,\mathcal{S}}\right]}{k} \\
&= \frac{\mathbb{E}\left[y_i \mathbf{A}_i^T \mathbf{x}\right]}{k} \\
&= \frac{\mathbb{E}\left[\mathbf{A}_i^T \mathbf{x}\, \mathbb{E}\left[y_1 | \mathbf{A}_i^T \mathbf{x}\right]\right]}{k} \\
&\overset{(c)}{=} \frac{\mathbb{E}\left[\mathbf{A}_i^T \mathbf{x}\, g\left(\mathbf{A}_i^T \mathbf{x}\right)\right]}{k} \\
&\overset{(d)}{=} \mathbb{E}\left[g'\left(\mathbf{A}_i^T \mathbf{x}\right)\right] := E \qquad (12)
\end{aligned}
$$

where $(a)$ follows from the fact that $y_i$ and $A_{i,j'}$ are zero mean, independent random variables and $(b)$ follows by defining $A_{i,\mathcal{S}} = \sum_{j \in \mathcal{S}} A_{i,j}$ and noticing that the random variables $y_i A_{i,j}$ are identically distributed for all $j \in \mathcal{S}$, $(c)$ follows from (6) and $(d)$ follows from Stein's lemma.

$$
\begin{aligned}
&\mathbb{P}\left(\sum_{i=1}^{m} (y_i(A_{i,j'} - A_{i,j})) \geq 0\right) \\
&= \mathbb{P}\left(\sum_{i=1}^{m} (y_i(A_{i,j} - A_{i,j'})) \leq 0\right) \\
&= \mathbb{P}\left(\sum_{i=1}^{m} (y_i(A_{i,j} - A_{i,j'})) - mE \leq -mE\right) \\
&\leq \mathbb{P}\left(\left|\sum_{i=1}^{m} (y_i(A_{i,j} - A_{i,j'})) - mE\right| \geq mE\right)
\end{aligned}
$$

To compute this, note that for all $i \in [1:m]$, $y_i$ is a subgaussian random variable and $y_i (A_{i,j} - A_{i,j'})$ being product of two subgaussian random variables is a subexponential random variable (see (Vershynin, 2018, Lemma 2.7.7)). Note that $\mathbb{E}\left[\sum_{i=1}^{m} (y_i(A_{i,j} - A_{i,j'}))\right] = mE$ where $E$ was defined in (12). Also,

$$\|y_i (A_{i,j} - A_{i,j'}) - E\|_{\psi_1}$$
$$\overset{(a)}{\leq} C \|y_i (A_{i,j} - A_{i,j'})\|_{\psi_1}$$
$$\overset{(b)}{\leq} C \|y_i\|_{\psi_2} \|(A_{i,j} - A_{i,j'})\|_{\psi_2}$$
$$\overset{(c)}{\leq} C \|y_i\|_{\psi_2} 2C'$$
$$= C_1 \|y_i\|_{\psi_2} \qquad \text{for some constant } C_1.$$

Here, $(a)$ follows from (Vershynin, 2018, Exercise 2.7.10), $(b)$ from (Vershynin, 2018, Lemma 2.7.7) and $(c)$ from (Vershynin, 2018, Example 2.5.8). With this

$$\mathbb{P}\left(\left|\sum_{i=1}^{m} (y_i(A_{i,j} - A_{i,j'})) - mE\right| \geq mE\right)$$
$$\overset{(a)}{\leq} 2\exp\left(-c\min\left(\frac{m^2 E^2}{mC_1^2 \|y_i\|_{\psi_2}^2}, \frac{mE}{C_1 \|y_i\|_{\psi_2}}\right)\right)$$
$$\overset{(b)}{\leq} 2\exp\left(-cm\min\left(\frac{mL^2}{C_1^2}, \frac{mL}{C_1}\right)\right)$$

where $(a)$ follows from (Vershynin, 2018, Theorem 2.8.1) and $(b)$ follows from the assumption in the lemma that $\frac{E}{\|y_i\|_{\psi_2}} = \frac{\mathbb{E}\left[g'(\mathbf{A}_i^T \mathbf{x})\right]}{\|y_i\|_{\psi_2}} \geq L$. Thus, from (10),

$$\mathbb{P}(\mathcal{F}^c) \leq k(n-k)2\exp\left(-C_2 m \min(L^2, L)\right)$$
$$\to 0 \text{ if } m \geq C_2 (\log k + \log(n-k)) \frac{1}{\min(L^2, L)}$$

for some constant $C_2$. □

*Proof of Theorem 2.5.* Suppose $\mathbf{x}$ is distributed uniformly on the set of all $k$-sparse binary vectors. Then,

$$I(\mathbf{A}, \mathbf{y}; \mathbf{x}) = H(\mathbf{x}) - H(\mathbf{x}|\mathbf{A}, \mathbf{y})$$
$$\overset{(a)}{\geq} \log\binom{n}{k} - h_2(\delta) - \delta\log\left(\binom{n}{k} - 1\right)$$
$$\geq k\log n/k - h_2(\delta) - \delta k\log(n) \qquad (13)$$

where $(a)$ follows from Fano's inequality (Cover & Thomas, 2006, Theorem 2.10.1). We also note that

$$I(\mathbf{A}, \mathbf{y}; \mathbf{x}) = I(\mathbf{A}; \mathbf{x}) + I(\mathbf{y}; \mathbf{x}|\mathbf{A})$$
$$\overset{(a)}{=} 0 + I(\mathbf{y}; \mathbf{x}|\mathbf{A}).$$

where $(a)$ holds because $\mathbf{A}$ and $\mathbf{x}$ are independent. Let $y_{j \in [1:i-1]}$ denote $(y_1, \ldots, y_{i-1})$.

$$I(\mathbf{y}; \mathbf{x}|\mathbf{A}) = \sum_{i=1}^{m} I(y_i; \mathbf{x}|\mathbf{A}, y_{j \in [1:i-1]})$$
$$= \sum_{i=1}^{m} \Big(H(y_i|\mathbf{A}, y_{j \in [1:i-1]})$$
$$\qquad - H(y_i|\mathbf{x}, \mathbf{A}, y_{j \in [1:i-1]})\Big)$$
$$\overset{(a)}{\leq} \sum_{i=1}^{m} (H(y_i|\mathbf{A}) - H(y_i|\mathbf{x}, \mathbf{A}))$$
$$= \sum_{i=1}^{m} I(y_i; \mathbf{x}|\mathbf{A})$$
$$\overset{(b)}{\leq} mI \qquad (14)$$

where $(a)$ follows from $H(y_i|\mathbf{A}, y_{j \in [1:i-1]}) \leq H(y_i|\mathbf{A})$ and $H(y_i|\mathbf{x}, \mathbf{A}, y_{j \in [1:i-1]}) = H(y_i|\mathbf{x}, \mathbf{A})$ as $y_i$ is conditionally independent of $y_{j \in [1:i-1]}$ conditioned on $\mathbf{x}$ and $\mathbf{A}$ and $(b)$ follows from the assumption in the Theorem. Thus, from (13) and (14),

$$mI \geq k\log(n/k)\left(1 - \frac{h_2(\delta) + \delta k\log(n)}{k\log n/k}\right)$$

This gives us the desired bound.

We can further simplify $I(y_i; \mathbf{x}|\mathbf{A})$ when $y_i \in \{-1, 1\}$,

$$I(y_i; \mathbf{x}|\mathbf{A}) = H(y_i|\mathbf{A}) - H(y_i|\mathbf{x}, \mathbf{A})$$
$$\overset{(a)}{\leq} 1 - H(y_i|\mathbf{x}, \mathbf{A}_i).$$

where $(a)$ holds because $H(y_i|\mathbf{A}) \leq H(y_i) = 1$ and $y_i$ is conditionally independent of $(\mathbf{A}_1 \ldots, \mathbf{A}_{i-1}, \mathbf{A}_{i+1}, \ldots, \mathbf{A}_m)$ conditioned on $\mathbf{A}_i$ and $\mathbf{x}$. Here $\mathbf{A}_i$, $i \in [1:m]$ denotes the $i^{\text{th}}$ row of the sensing matrix $\mathbf{A}$.

Suppose $\mathbf{x}$ is fixed and $\mathbb{P}(y_i = 1) = \frac{1}{2} + t$ for some $t \in [-1/2, 1/2]$. Then $\mathbb{E}[y_i|\mathbf{A}_i] = 2t = g(\mathbf{A}_i^T \mathbf{x})$.

$$H(y_i|\mathbf{A}_i, \mathbf{x}) \overset{(a)}{=} \mathbb{E}\left[h_2\left(\frac{1}{2} + t\right)\right]$$
$$\overset{(b)}{\geq} \mathbb{E}_{\mathbf{x}}\left[\mathbb{E}_{\mathbf{A}}\left[4\left(\frac{1}{2} + t\right)\left(\frac{1}{2} - t\right)\Big|\mathbf{x}\right]\right]$$
$$= 1 - \mathbb{E}_{\mathbf{x}}\left[\mathbb{E}\left[(2t)^2\Big|\mathbf{x}\right]\right]$$
$$= 1 - \mathbb{E}_{\mathbf{x}}\left[\mathbb{E}\left[\left(g(\mathbf{A}_i^T \mathbf{x})\right)^2\Big|\mathbf{x}\right]\right]$$
$$= 1 - \mathbb{E}_{\mathbf{A}, \mathbf{x}}\left[\left(g(\mathbf{A}_i^T \mathbf{x})\right)^2\right]$$

where in $(a)$, the expectation is over $\mathbf{A}$ and $\mathbf{x}$. The inequality

($b$) follows from (Topsøe, 2001, Theorem 1.2)[4]. With this
$$I(y_i; \mathbf{x}|\mathbf{A}) \leq \mathbb{E}\left[\left(g(\mathbf{A}_i^T \mathbf{x})\right)^2\right]. \qquad \square$$

## 4. Conclusion and open problems

We analyze a simple algorithm (the "average algorithm" from (Plan et al., 2017) followed by 'top-k' selection) for recovering sparse binary vectors from generalized linear measurements; along with an information theoretic lower bound. This gives optimal sample complexity characterization for 1bCSbinary and LogisticRegression. On the other hand, the required number of measurements for the noisy linear case (SparseLinearReg), which is $O((k + \sigma^2) \log n)$, is as good as the sample complexity of any other known efficient algorithm for this problem, up to constants. An interesting open problem is to find a design matrix and an efficient algorithm which requires less than $(k + \sigma^2) \log n$ samples for SparseLinearReg. When the noise variance is zero, we show such an algorithm in Remark 1.1.

We also present almost matching information theoretic upper and lower bounds for SparseLinearReg given by (8) and (9) respectively. The bounds are in the form of an optimization problem. While we present numerical evidence which suggests that (8) is optimized by $l = k\left(1 - \frac{k}{n}\right)$, a formal proof is still missing. The bounds in (8) and (9) also differ slightly by constants in the denominator, which seems to be a persistent gap in this problem.

**Acknowledgment**   This work is supported in part by NSF awards 2217058 and 2112665. The authors would like to thank Krishna Narayanan who introduced them to the binary linear regression problem at the Simons Institute program on Error-correcting codes.

## Impact Statement

This paper presents work whose goal is to advance the field of Machine Learning. There are many potential societal consequences of our work, none which we feel must be specifically highlighted here.

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

## A. Proofs

### A.1. Missing proofs from Section 2.1

*Proof of Corollary 2.2.* We need to compute a lower bound $L$ on $\frac{\mathbb{E}[g'(\mathbf{A}_i^T\mathbf{x})]}{\|y_i\|_{\psi_2}}$. Instead of computing $\mathbb{E}\left[g'(\mathbf{A}_i^T\mathbf{x})\right]$, we will compute $\mathbb{E}\left[y_i A_{i,j}\right]$ for any $j$ in the support of $\mathbf{x}$. From (11) and (12), we note that $\mathbb{E}\left[y_i A_{i,j}\right] = \mathbb{E}\left[g'(\mathbf{A}_i^T\mathbf{x})\right]$. Also note that $\mathbb{E}\left[y_i A_{i,j}\right] = \mathbb{E}\left[A_{i,j}\mathbb{E}\left[y_i|A_{i,j}\right]\right]$.

For any $\mathcal{U} \subseteq [1:n]$, we denote $\sum_{l \in \mathcal{U}} A_{i,l}$ by $A_{i,\mathcal{U}}$. For any $A_{i,j} = a$,

$$
\begin{aligned}
&\mathbb{P}\left(y_i = 1|A_{i,j} = a\right) \\
&= \mathbb{P}\left(A_{i,\mathcal{S}\setminus\{j\}} + z_i \geq -a\right) \\
&= \mathbb{P}\left(\frac{A_{i,\mathcal{S}\setminus\{j\}} + z_i}{\sqrt{k-1+\sigma^2}} \geq -\frac{a}{\sqrt{k-1+\sigma^2}}\right) \\
&= 1 - \Phi\left(-\frac{a}{\sqrt{k-1+\sigma^2}}\right)
\end{aligned}
$$

where $\Phi(x) = \frac{1}{2\pi}\int_{-\infty}^x e^{-\frac{t^2}{2}}\,dt$ is the cumulative distribution function of the standard Gaussian distribution. Thus, $\mathbb{P}\left(y_i = -1|A_{i,j} = a\right) = \Phi\left(-\frac{a}{\sqrt{k-1+\sigma^2}}\right)$ and

$$
\mathbb{E}\left[y_i|A_{i,j} = a\right] = 1 - 2\Phi\left(-\frac{a}{\sqrt{k-1+\sigma^2}}\right).
$$

We are now ready to compute $\mathbb{E}\left[A_{i,j}\mathbb{E}\left[y_i|A_{i,j}\right]\right]$.

$$
\begin{aligned}
&\mathbb{E}\left[A_{i,j}\mathbb{E}\left[y_i|A_{i,j}\right]\right] \\
&= \mathbb{E}\left[A_{i,j}\left(1 - 2\Phi\left(-\frac{A_{i,j}}{\sqrt{k-1+\sigma^2}}\right)\right)\right] \\
&= \mathbb{E}\left[A_{i,j}\right] - 2\mathbb{E}\left[A_{i,j}\Phi\left(-\frac{A_{i,j}}{\sqrt{k-1+\sigma^2}}\right)\right] \\
&= 0 - 2\mathbb{E}\left[A_{i,j}\Phi\left(-\frac{A_{i,j}}{\sqrt{k-1+\sigma^2}}\right)\right]
\end{aligned}
\tag{15}
$$

$$
\begin{aligned}
&\mathbb{E}\left[A_{i,j}\Phi\left(-\frac{A_{i,j}}{\sqrt{k-1+\sigma^2}}\right)\right] \\
&= \int_{-\infty}^\infty a\frac{1}{\sqrt{2\pi}}e^{-\frac{a^2}{2}}\left(\frac{1}{\sqrt{2\pi}}\int_{-\infty}^{-\frac{a}{\sqrt{k-1+\sigma^2}}} e^{-\frac{t^2}{2}}\,dt\right)da \\
&= \frac{1}{2\pi}\int_{-\infty}^\infty \int_{-\infty}^{-\frac{a}{\sqrt{k-1+\sigma^2}}} ae^{-\frac{a^2}{2}}e^{-\frac{t^2}{2}}\,dt\,da \\
&\overset{(a)}{=} \frac{1}{2\pi}\int_{-\infty}^\infty \int_{-\infty}^{-t\sqrt{k-1+\sigma^2}} ae^{-\frac{a^2}{2}}e^{-\frac{t^2}{2}}\,da\,dt \\
&= \frac{1}{2\pi}\int_{-\infty}^\infty \left(\int_{-\infty}^{-t\sqrt{k-1+\sigma^2}} ae^{-\frac{a^2}{2}}\,da\right)e^{-\frac{t^2}{2}}\,dt \\
&= \frac{1}{2\pi}\int_{-\infty}^\infty \left(-e^{-\frac{t^2(k-1+\sigma^2)}{2}}\right)e^{-\frac{t^2}{2}}\,dt \\
&= -\frac{1}{\sqrt{2\pi(k+\sigma^2)}}\int_{-\infty}^\infty \frac{\sqrt{k+\sigma^2}}{\sqrt{2\pi}}e^{-\frac{t^2(k+\sigma^2)}{2}}\,dt \\
&= -\frac{1}{\sqrt{2\pi(k+\sigma^2)}}
\end{aligned}
\tag{16}
$$

where $(a)$ follows for change of variable formula for integration. From (15) and (16), we have

$$\mathbb{E}\left[y_i A_{i,j}\right] = \sqrt{\frac{2}{\pi}} \times \frac{1}{\sqrt{(k+\sigma^2)}}. \tag{17}$$

From (Vershynin, 2018, Example 2.5.8), we also note that $\|y_i\|_{\psi_2} = 1$. Thus, $L = \sqrt{\frac{2}{\pi}} \times \frac{1}{\sqrt{(k+\sigma^2)}}$ and $\min\left\{L, L^2\right\} = L^2$, which when substituted in (7) gives the desired bound. $\qquad\square$

*Proof of Corollary 2.3.* We first note that $\frac{\mathbb{E}\left[g'(\mathbf{A}_i^T \mathbf{x})\right]}{\|y_i\|_{\psi_2}} = \frac{\mathbb{E}\left[y_i A_{i,j}\right]}{\|y_i\|_{\psi_2}}$ for any $j$ in the support of $\mathbf{x}$. This follows from (11) and (12). We first compute $\mathbb{E}\left[y_i A_{i,j}\right]$, which is the same as $\mathbb{E}\left[\left(\mathbf{A}_i^T \mathbf{x} + z_i\right) A_{i,j}\right]$ for SparseLinearReg. Note that $\mathbb{E}\left[\left(\mathbf{A}_i^T \mathbf{x} + z_i\right)(A_{i,j})\right] = \mathbb{E}\left[A_{i,j}^2\right] = 1$. Also, from (Vershynin, 2018, Example 2.5.8)

$$\left\|\left(\mathbf{A}_i^T \mathbf{x} + z_i\right)\right\|_{\psi_2} \leq C\sqrt{k + \sigma^2}$$

for some constants $C$. With this,

$$\frac{\mathbb{E}\left[g'(\mathbf{A}_i^T \mathbf{x})\right]}{\|y_i\|_{\psi_2}} \geq \frac{1}{C\sqrt{k + \sigma^2}} := L.$$

Note that $\min\left\{L, L^2\right\} = L^2$, which when substituted in (7) gives the desired bound. $\qquad\square$

*Proof of Corollary 2.4.* We will first compute $g(\mathbf{A}_i^T \mathbf{x}) = \mathbb{E}\left[y_i | \mathbf{A}_i^T \mathbf{x}\right]$ for LogisticRegression.

$$
\begin{aligned}
g(\mathbf{A}_i^T \mathbf{x}) &= \mathbb{E}\left[y_i | \mathbf{A}_i^T \mathbf{x}\right] \\
&= \frac{1}{1 + e^{-\beta \mathbf{A}_i^T \mathbf{x}}} - \frac{e^{-\beta \mathbf{A}_i^T \mathbf{x}}}{1 + e^{-\beta \mathbf{A}_i^T \mathbf{x}}} \\
&= \frac{1 - e^{-\beta \mathbf{A}_i^T \mathbf{x}}}{1 + e^{-\beta \mathbf{A}_i^T \mathbf{x}}} \\
&\overset{(a)}{=} \tanh\left(\frac{\beta \mathbf{A}_i^T \mathbf{x}}{2}\right)
\end{aligned}
$$

where $(a)$ uses the definition of $\tanh$. Then

$$
\begin{aligned}
\mathbb{E}\left[g'(\mathbf{A}_i^T \mathbf{x})\right] &= \frac{\beta}{2}\mathbb{E}\left[\frac{1}{\cosh^2\left(\frac{\beta \mathbf{A}_i^T \mathbf{x}}{2}\right)}\right] \\
&\overset{(c)}{\geq} \frac{\beta}{2}\mathbb{E}\left[e^{-\frac{\left(\beta \mathbf{A}_i^T \mathbf{x}\right)^2}{4}}\right]
\end{aligned}
$$

where $(c)$ follows from the inequality $\cosh(t) \leq e^{t^2/2}$ (see (Vershynin, 2018, Exercise 2.2.3)).

Now, we need to compute $\mathbb{E}\left[e^{-\frac{\left(\beta \mathbf{A}_i^T \mathbf{x}\right)^2}{4}}\right]$ where $\mathbf{A}_i^T \mathbf{x} \sim N(0, k)$. Let $\sigma_1 := \frac{1}{\frac{\beta^2}{2} + \frac{1}{k}}$. Then

$$\mathbb{E}\left[e^{-\frac{\left(\beta \mathbf{A}_i^T \mathbf{x}\right)^2}{4}}\right] = \int_{-\infty}^{\infty} \frac{1}{\sqrt{2\pi k}} e^{-\beta^2 a^2/4} e^{-a^2/2k} da$$

$$= \sqrt{\frac{\sigma_1}{k}} \int_{-\infty}^{\infty} \frac{1}{\sqrt{2\pi\sigma_1}} e^{-x^2/2\sigma_1} da$$

$$= \sqrt{\frac{\sigma_1}{k}}$$

$$= \sqrt{\frac{2}{2 + \beta^2 k}} \tag{18}$$

Thus,

$$\mathbb{E}\left[g'(\mathbf{A}_i^T \mathbf{x})\right] \geq \frac{\beta}{2} \sqrt{\frac{2}{2 + \beta^2 k}}$$

$$= \frac{1}{2} \sqrt{\frac{2}{2/\beta^2 + k}}.$$

From (Vershynin, 2018, Example 2.5.8), we also note that $\|y_i\|_{\psi_2} = 1$. Thus, $L = \frac{1}{2}\sqrt{\frac{2}{2/\beta^2 + k}}$ and $\min\left(L, L^2\right) = L^2$, which gives the desired bound.

$\square$

## A.2. Missing proofs from Section 2.2

*Proof of Corollary 2.6.* Consider a sensing matrix $\mathbf{A}$ which satisfies the power constraint (2).

Here $\mathbf{A}_i$, $i \in [1 : m]$ denotes the $i^{\text{th}}$ row of the sensing matrix $\mathbf{A}$. Let $Q(\cdot)$ be the Gaussian $Q$ function. For any realization $b \in \mathbb{R}$ of $\mathbf{A}_i^T \mathbf{x}$,

$$\mathbb{P}(y_i = 1 | \mathbf{A}_i^T \mathbf{x} = b) = \mathbb{P}(z_i \geq -b) = \mathbb{P}\left(\frac{z_i}{\sigma} \geq \frac{-b}{\sigma}\right)$$

$$= \frac{1 - \text{sign}(b)}{2} + \text{sign}(b)Q\left(\frac{|b|}{\sigma}\right).$$

For $a > 0$, let $R(a) := \frac{1}{\sqrt{2\pi}} \int_0^a e^{-u^2/2} du$. Then $Q(a) = \frac{1}{2} - R(a)$. Suppose $\mathbf{x}$ is fixed. Then,

$$g(\mathbf{A}_i^T \mathbf{x}) = \mathbb{E}[y_i | \mathbf{A}] = \mathbb{E}\left[y_i | \mathbf{A}_i^T \mathbf{x}\right]$$

$$= \frac{1 - \text{sign}(\mathbf{A}_i^T \mathbf{x})}{2} + \text{sign}(\mathbf{A}_i^T \mathbf{x})Q\left(\frac{|\mathbf{A}_i^T \mathbf{x}|}{\sigma}\right) - \left(1 - \left(\frac{1 - \text{sign}(\mathbf{A}_i^T \mathbf{x})}{2} + \text{sign}(\mathbf{A}_i^T \mathbf{x})Q\left(\frac{|\mathbf{A}_i^T \mathbf{x}|}{\sigma}\right)\right)\right)$$

$$= \text{sign}(\mathbf{A}_i^T \mathbf{x})\left(1 - 2Q\left(\frac{|\mathbf{A}_i^T \mathbf{x}|}{\sigma}\right)\right)$$

$$= \text{sign}(\mathbf{A}_i^T \mathbf{x})\left(2R\left(\frac{|\mathbf{A}_i^T \mathbf{x}|}{\sigma}\right)\right)$$

For any $a > 0$,

$$R(a) = \frac{1}{\sqrt{2\pi}} \int_0^a e^{-u^2/2} du$$

$$\leq \frac{1}{\sqrt{2\pi}} \int_0^a 1 du = \frac{a}{\sqrt{2\pi}}.$$

Thus,

$$
\mathbb{E}\left[\left(g\left(\mathbf{A}_i^T\mathbf{x}\right)^2\right)\right] = \mathbb{E}\left[\left(2R\left(\frac{|\mathbf{A}_i^T\mathbf{x}|}{\sigma}\right)\right)^2\right]
$$

$$
\leq \mathbb{E}\left[4\left(\frac{\mathbf{A}_i^T\mathbf{x}}{\sqrt{2\pi}\sigma}\right)^2\right]
$$

$$
\overset{(a)}{\leq} \frac{2k}{\pi\sigma^2}
$$

where $(a)$ follows from the power constraint $\mathbb{E}\left[\left(\mathbf{A}_i^T\mathbf{x}\right)^2\right] \leq k$ (see (2)). This holds for any $\mathbf{x}$, including a randomly chosen sparse vector. Thus,

$$
m \geq \frac{\pi\sigma^2}{2k}k\log(n/k)\left(1 - \frac{h_2(\delta) + \delta k\log(n)}{k\log n/k}\right)
$$

$$
\geq \sigma^2\log(n/k)\left(1 - \frac{h_2(\delta) + \delta k\log(n)}{k\log n/k}\right) \tag{19}
$$

On the other hand, $I(y_i; \mathbf{x}|\mathbf{A}) \leq 1$. Thus,

$$
k\log(n/k)\left(1 - \frac{h_2(\delta) + \delta k\log(n)}{k\log n/k}\right)
$$

$$
\leq \sum_{i=1}^m I(y_i; \mathbf{x}|\mathbf{A}) \leq \sum_{i=1}^m H(y_i|\mathbf{A})
$$

$$
\leq m. \tag{20}
$$

Combining (19) and (20), we get the desired bound. □

*Proof of Corollary 2.7.* Consider a Gaussian sensing matrix $\mathbf{A}$. Suppose $\mathbf{x}$ is distributed uniformly on the set of all $k$-sparse binary vectors.

Suppose $t = \frac{1}{2}\tanh\frac{\beta\mathbf{A}_i^T\mathbf{x}}{2}\left(= \frac{(1-e^{-\beta\mathbf{A}_i^T\mathbf{x}})}{2(1+e^{-\beta\mathbf{A}_i^T\mathbf{x}})}\right)$. Then,

$$
\frac{1}{1 + e^{-\beta\mathbf{A}_i^T\mathbf{x}}} = \frac{1}{2} + t \text{ and}
$$

$$
1 - \frac{1}{1 + e^{-\beta\mathbf{A}_i^T\mathbf{x}}} = \frac{1}{2} - t
$$

With this,

$$
\mathbb{E}\left[\left(g\left(\mathbf{A}_i^T\mathbf{x}\right)^2\right)\right] = \mathbb{E}\left[4t^2\right]
$$

$$
= \mathbb{E}\left[\left(\tanh\frac{\beta\mathbf{A}_i^T\mathbf{x}}{2}\right)^2\right]
$$

Note that,

$$
\mathbb{E}\left[\left(\tanh\frac{\beta\mathbf{A}_i^T\mathbf{x}}{2}\right)^2\right] = 1 - \mathbb{E}\left[\left(\text{sech}\frac{\beta\mathbf{A}_i^T\mathbf{x}}{2}\right)^2\right]
$$

and

$$\mathbb{E}\left[\left(\operatorname{sech}\frac{\beta \mathbf{A}_i^T \mathbf{x}}{2}\right)^2\right] = \mathbb{E}\left[\frac{1}{\left(\cosh\frac{\beta \mathbf{A}_i^T \mathbf{x}}{2}\right)^2}\right]$$

$$\overset{(a)}{\geq} \mathbb{E}\left[e^{-\left(\beta \mathbf{A}_i^T \mathbf{x}/2\right)^2}\right]$$

$$\overset{(b)}{=} \sqrt{\frac{1}{1+\beta^2 k/2}}$$

$$\overset{(c)}{\geq} 1 - \frac{\beta^2 k}{2}$$

where $(a)$ follows from the inequality $\cosh(t) \leq e^{t^2/2}$ (see (Vershynin, 2018, Exercise 2.2.3)), $(b)$ follows from (18) and $(c)$ holds because $1 - \frac{x}{2} \leq \frac{1}{\sqrt{1+x}}$ for any $x \geq 0$. Thus,

$$\mathbb{E}\left[\left(g\left(\mathbf{A}_i^T \mathbf{x}\right)^2\right)\right] \leq \frac{\beta^2 k}{2}.$$

This implies that

$$m\frac{\beta^2 k}{2} \geq k \log\left(n/k\right)\left(1 - \frac{h_2(\delta) + \delta k \log\left(n\right)}{k \log n/k}\right)$$

Thus,

$$m \geq \frac{2}{\beta^2} \log\left(n/k\right)\left(1 - \frac{h_2(\delta) + \delta k \log\left(n\right)}{k \log n/k}\right)$$

$$\geq \frac{1}{\beta^2} \log\left(n/k\right)\left(1 - \frac{h_2(\delta) + \delta k \log\left(n\right)}{k \log n/k}\right) \tag{21}$$

We also know that for any $i$, $I(y_i; \mathbf{x}|\mathbf{A}) \leq H(y_i|\mathbf{A}) \leq 1$. Thus, we also obtain that

$$m \geq k \log\left(n/k\right)\left(1 - \frac{h_2(\delta) + \delta k \log\left(n\right)}{k \log n/k}\right) \tag{22}$$

Combining (21) and (22), we get the desired bound. □

*Proof of Corollary 2.8.* Suppose $\mathbf{x}$ is generated uniformly at random from the set of all $k$-sparse vectors and $\mathbf{A}$ is any sensing matrix which satisfies the power constraint given by (2). Then,

$$I(y_i; \mathbf{x}|\mathbf{A}) = h(y_i|\mathbf{A}) - h(y_i|\mathbf{x}, \mathbf{A})$$
$$\leq \left(h(y_i) - h(\mathbf{A}_i^T \mathbf{x} + z_i|\mathbf{x}, \mathbf{A})\right)$$
$$= h(y_i) - h(z_i)$$
$$\leq \left(h(w_i) - h(z_i)\right)$$

where in the last inequality, $w_i \sim \mathcal{N}\left(0, \sigma_w^2\right)$ where $\mathsf{Var}(y_i) \leq \sigma_w^2$. We will now compute an upper bound on $\mathsf{Var}(y_i)$.

$$\mathsf{Var}(y_i) \leq \mathbb{E}\left[\left(\mathbf{A}_i^T \mathbf{x} + z_i\right)^2\right] = \mathbb{E}\left[\left(\mathbf{A}_i^T \mathbf{x}\right)^2\right] + \sigma^2$$
$$\leq k + \sigma^2$$

Thus, we have

$$\left(h(w_i) - h(z_i)\right) = \frac{1}{2} \log\left(\frac{k}{\sigma^2} + 1\right). \tag{23}$$

With this, we conclude that

$$m \geq \frac{2k \log \left(\frac{n}{k}\right) - h_2(\delta) - \delta k \log n}{\frac{1}{2} \log \left(\frac{k}{\sigma^2} + 1\right)}.$$

$\square$

### A.3. Missing proofs from Section 2.3

*Proof of Theorem 2.9.* We consider a sensing matrix $\mathbf{A}$ where each entry is chosen iid $\mathcal{N}(0,1)$. let $\mathcal{X}_k$ denote the set of all $k$-sparse binary vectors. That is $\mathcal{X}_k = \{\mathbf{x}' \in \{0,1\}^n : |\mathbf{x}|_{\mathsf{H}} = k\}$. We decode to $\hat{\mathbf{x}}$ if, on output $\mathbf{y}$,

$$\hat{\mathbf{x}} = \arg\max_{\mathbf{x}' \in \mathcal{X}_k} \ p(\mathbf{y}|\mathbf{x}')$$

where $p(\mathbf{y}|\mathbf{x}')$ is the probability density function of $\mathbf{y}$ on input $\mathbf{x}'$. We assume that $k \leq n/2$. Suppose unknown signal is $\mathbf{x}$. The error event $\mathcal{E}$ is

$$\mathcal{E} = \{\mathbf{y} : \exists \tilde{\mathbf{x}} \neq \mathbf{x} \text{ such that } p(\mathbf{y}|\tilde{\mathbf{x}}) > p(\mathbf{y}|\mathbf{x})\}$$

Then

$$\mathbb{P}(\mathcal{E}) \leq \sum_{l=1}^{k} \sum_{\substack{\tilde{\mathbf{x}} \in \mathcal{X}_k: \\ \mathsf{d}_{\mathsf{H}}(\mathbf{x},\tilde{\mathbf{x}})=2l}} \mathbb{P}\left(p(\mathbf{y}|\tilde{\mathbf{x}}) > p(\mathbf{y}|\mathbf{x})\right)$$

Suppose $\mathbf{x}$ has support on $\mathcal{S} \subset [1:n], |\mathcal{S}| = k$ and $\tilde{\mathbf{x}}$ has support on $\mathcal{U} \subset [1:n], |\mathcal{U}| = k$. Then, conditioned on $\mathbf{x}$, $y_r$ is generated from $\sum_{i \in \mathcal{S}} A_{r,i}$ which we denote by $A_{r,\mathcal{S}}$. That is, $y_r = A_{r,\mathcal{S}} + z_r$ where $A_{r,\mathcal{S}} \sim \mathcal{N}(0,k)$. Similarly, conditioned on $\tilde{\mathbf{x}}$, $y_r = A_{r,\mathcal{U}} + z_r$ for $A_{r,\mathcal{U}} := \sum_{i \in \mathcal{U}} A_{r,i}$ where $A_{r,\mathcal{U}} \sim \mathcal{N}(0,k)$. For any $l \in [1:k]$, computing $\mathbb{P}\left(p(\mathbf{y}|\tilde{\mathbf{x}}) > p(\mathbf{y}|\mathbf{x})\right)$, we have

$$\mathbb{P}(p(\mathbf{y}|\tilde{\mathbf{x}}) > p(\mathbf{y}|\mathbf{x}))$$
$$= \mathbb{P}\left(\log \frac{p(\mathbf{y}|\tilde{\mathbf{x}})}{p(\mathbf{y}|\mathbf{x})} > 0\right)$$
$$= \mathbb{P}\left(\sum_{r=1}^{m} \log \frac{p(y_r|A_{r,\mathcal{U}})}{p(y_r|A_{r,\mathcal{S}})} > 0\right)$$
$$\stackrel{(a)}{=} \mathbb{P}\left(\sum_{r=1}^{m} -\frac{(y_r - A_{r,\mathcal{U}})^2}{2\sigma^2} + \frac{(y_r - A_{r,\mathcal{S}})^2}{2\sigma^2} > 0\right)$$
$$= \mathbb{P}\left(\sum_{r=1}^{m}(A_{r,\mathcal{U}} - A_{r,\mathcal{S}})y_r > \right.$$
$$\left. \sum_{r=1}^{m}(A_{r,\mathcal{U}} - A_{r,\mathcal{S}})\frac{(A_{r,\mathcal{U}} + A_{r,\mathcal{S}})}{2}\right)$$

where in $(a)$, we used the Gaussian density formula which states that for any $r$ and $\mathcal{V}$, $p(y_r|A_{r,\mathcal{V}}) = \frac{1}{\sqrt{2\pi\sigma^2}}e^{-\frac{\left(y_r - A_{r,\mathcal{V}}\right)^2}{2\sigma^2}}$.
Using the fact that $y_r = A_{r,\mathcal{S}\setminus\mathcal{U}} + A_{r,\mathcal{S}\cap\mathcal{U}} + z_r$, we have

$$
\begin{aligned}
&\mathbb{P}\left(p(\mathbf{y}|\tilde{\mathbf{x}}) > p(\mathbf{y}|\mathbf{x})\right) \\
&= \mathbb{P}\Big(\sum_{r=1}^{m}(A_{r,\mathcal{U}\setminus\mathcal{S}} - A_{r,\mathcal{S}\setminus\mathcal{U}})(A_{r,\mathcal{S}\setminus\mathcal{U}} + A_{r,\mathcal{S}\cap\mathcal{U}} + z_r) > \\
&\qquad \sum_{r=1}^{m}(A_{r,\mathcal{U}\setminus\mathcal{S}} - A_{r,\mathcal{S}\setminus\mathcal{U}})\frac{A_{r,\mathcal{U}\setminus\mathcal{S}} + A_{r,\mathcal{S}\setminus\mathcal{U}} + 2A_{r,\mathcal{U}\cap\mathcal{S}}}{2}\Big) \\
&= \mathbb{P}\Big(\sum_{r=1}^{m}(A_{r,\mathcal{U}\setminus\mathcal{S}} - A_{r,\mathcal{S}\setminus\mathcal{U}})z_r > \\
&\qquad \sum_{r=1}^{m}\frac{A_{r,\mathcal{U}\setminus\mathcal{S}}^2}{2} - \frac{A_{r,\mathcal{S}\setminus\mathcal{U}}^2}{2} - A_{r,\mathcal{U}\setminus\mathcal{S}}A_{r,\mathcal{S}\setminus\mathcal{U}} + A_{r,\mathcal{S}\setminus\mathcal{U}}^2\Big) \\
&= \mathbb{P}\Bigg(\sum_{r=1}^{m}(A_{r,\mathcal{U}\setminus\mathcal{S}} - A_{r,\mathcal{S}\setminus\mathcal{U}})z_r > \\
&\qquad \frac{\sum_{r=1}^{m}\left(A_{r,\mathcal{U}\setminus\mathcal{S}} - A_{r,\mathcal{S}\setminus\mathcal{U}}\right)^2}{2}\Bigg) \\
&= \mathbb{P}\Bigg(\frac{\sum_{r=1}^{m}(A_{r,\mathcal{U}\setminus\mathcal{S}} - A_{r,\mathcal{S}\setminus\mathcal{U}})z_r}{\sqrt{\sum_{r=1}^{m}\left(A_{r,\mathcal{U}\setminus\mathcal{S}} - A_{r,\mathcal{S}\setminus\mathcal{U}}\right)^2}\sigma} \\
&\qquad > \frac{\sqrt{\sum_{r=1}^{m}\left(A_{r,\mathcal{U}\setminus\mathcal{S}} - A_{r,\mathcal{S}\setminus\mathcal{U}}\right)^2}}{2\sigma}\Bigg).
\end{aligned}
$$

Let $b_r = A_{r,\mathcal{U}\setminus\mathcal{S}} - A_{r,\mathcal{S}\setminus\mathcal{U}}$. Note that $b_r \sim \mathcal{N}(0, 2l)$. Let $\mathbf{b} = (b_1, \ldots, b_m)$. Let $\mathbf{e} = (e_1, \ldots, e_m)$ denote the realization of $\mathbf{b}$. Then,

$$
\begin{aligned}
&\mathbb{P}\left(\frac{\sum_{r=1}^{m}(A_{r,\mathcal{U}\setminus\mathcal{S}} - A_{r,\mathcal{S}\setminus\mathcal{U}})z_r}{\sqrt{\sum_{r=1}^{m}\left(A_{r,\mathcal{U}\setminus\mathcal{S}} - A_{r,\mathcal{S}\setminus\mathcal{U}}\right)^2}\sigma} > \right. \\
&\qquad\qquad \left.\frac{\sqrt{\sum_{r=1}^{m}\left(A_{r,\mathcal{U}\setminus\mathcal{S}} - A_{r,\mathcal{S}\setminus\mathcal{U}}\right)^2}}{2\sigma}\right) \\
&= \mathbb{P}\left(\frac{\sum_{r=1}^{m}b_r z_r}{\sqrt{\sum_{r=1}^{m}(b_r)^2}\sigma} > \frac{\sqrt{\sum_{r=1}^{m}(b_r)^2}}{2\sigma}\right) \\
&\overset{(a)}{=} \int p_{\mathbf{b}}(\mathbf{e})\mathbb{P}\left(\frac{\sum_{r=1}^{m}e_r z_r}{\sqrt{\sum_{r=1}^{m}(e_r)^2}\sigma} > \frac{\sqrt{\sum_{r=1}^{m}(e_r)^2}}{2\sigma}\right)d\mathbf{e} \\
&= \int p_{\mathbf{b}}(\mathbf{e})Q\left(\frac{\sqrt{\sum_{r=1}^{m}(e_r)^2}}{2\sigma}\right)d\mathbf{e}
\end{aligned}
$$

where in $(a)$, $p_\mathbf{b}(\mathbf{e})$ denotes the density of $\mathbf{b}$ at $\mathbf{e}$ and $d\mathbf{e}$ is shorthand for $de_1 de_2 \ldots de_m$. To analyse this further, we use the upper bound $Q(x) \leq \frac{1}{2} e^{-x^2/2}$.

$$\int p_\mathbf{b}(\mathbf{e}) Q\left(\frac{\sqrt{\sum_{r=1}^{m}(e_r)^2}}{2\sigma}\right) d\mathbf{e}$$

$$= \int \frac{1}{(2\pi \cdot 2l)^{m/2}} 2^{\left(-\frac{\sum_{r=1}^{m} e_r^2}{2l}\right)} 2^{\left(-\frac{\sum_{r=1}^{m} e_r^2}{8\sigma^2}\right)} d\mathbf{e}$$

$$= \int \frac{1}{(2\pi \cdot 2l)^{m/2}} 2^{\left(-\sum_{r=1}^{m} e_r^2 \left(\frac{1}{2l} + \frac{1}{8\sigma^2}\right)\right)} d\mathbf{e}$$

$$= \frac{1}{\left(\frac{1}{2l} + \frac{1}{8\sigma^2}\right)^{m/2} (2l)^{m/2}}$$

$$\int \frac{1}{(2\pi)^{m/2}} \left(\frac{1}{2l} + \frac{1}{8\sigma^2}\right)^{m/2} 2^{\left(-\sum_{r=1}^{m} e_r^2 \left(\frac{1}{2l} + \frac{1}{8\sigma^2}\right)\right)} d\mathbf{e}$$

$$= \left(\frac{1}{1 + \frac{l}{2\sigma^2}}\right)^{m/2}$$

$$= 2^{\left(-\frac{m}{2} \log\left(1 + \frac{l}{2\sigma^2}\right)\right)}$$

Next, we observe that

$$|\{\mathbf{x}' \in \mathcal{X}_k : \mathsf{d}_\mathsf{H}(\mathbf{x}, \mathbf{x}') = 2l\}| = \binom{k}{l}\binom{n-k}{l}$$

$$\overset{(a)}{\leq} 2^{kh_2\left(\frac{l}{k}\right)} 2^{(n-k)h_2\left(\frac{l}{n-k}\right)}$$

$$= 2^{n\left(\frac{k}{n} h_2\left(\frac{l}{k}\right) + \frac{(n-k)}{n} h_2\left(\frac{l}{(n-k)}\right)\right)}$$

$$= 2^{nN(l)}.$$

where $(a)$ uses the inequality $\binom{n}{k} \leq 2^{nh_2(k/n)}$ ((Cover & Thomas, 2006, Theorem 11.1.3)). Then,

$$\mathbb{P}(\mathcal{E}) \leq \sum_{l=1}^{k} 2^{nN(l)} 2^{\left(-\frac{m}{2} \log\left(1 + \frac{l}{2\sigma^2}\right)\right)}$$

$$\to 0 \text{ if } m \geq \max_l \frac{2nN(l)}{\log\left(1 + \frac{l}{2\sigma^2}\right)}$$

$\square$

*Proof of Theorem 2.10.* We consider a joint distribution given by the following process. $\tilde{\mathbf{x}}$ is generated uniformly at random from the set of all $k$-sparse vectors. Given $\tilde{\mathbf{x}}$, the unknown signal $\mathbf{x}$ is chosen uniformly at random from the set of all vectors which are at a Hamming distance $2l$ from $\tilde{\mathbf{x}}$ for some $l \in [1 : k]$ (assuming $k \leq n/2$). We will denote the realization of $\tilde{\mathbf{x}}$ by $\bar{\mathbf{x}}$ and the realization of $\mathbf{x}$ by $\hat{\mathbf{x}}$. With this, given $\tilde{\mathbf{x}} = \bar{\mathbf{x}}$,

$$\mathbb{P}\left(\mathbf{x} = \hat{\mathbf{x}} | \tilde{\mathbf{x}} = \bar{\mathbf{x}}\right) = \frac{1}{\binom{k}{l}\binom{n-k}{l}}.$$

Note that the marginal distribution of $\mathbf{x}$ is uniform over the set of all $k$-sparse vectors.

We will be using the below set of equations in our further analysis. For $\mathbf{x} = (x_1, \ldots, x_n)$, any $j, l \in \mathcal{S}_{\bar{\mathbf{x}}}$ where $j \neq l$, we

have

$$\mathbb{P}\left(x_j = 1 | \tilde{\mathbf{x}} = \bar{\mathbf{x}}\right) = \frac{\binom{k-1}{l}\binom{n-k}{l}}{\binom{k}{l}\binom{n-k}{l}} = \frac{k-l}{k}, \text{ and} \tag{24}$$

$$\mathbb{P}\left(x_j = x_l = 1 | \tilde{\mathbf{x}} = \bar{\mathbf{x}}\right) = \frac{\binom{k-2}{l}\binom{n-k}{l}}{\binom{k}{l}\binom{n-k}{l}} = \left(\frac{k-l}{k}\right)\left(\frac{k-l-1}{k-1}\right), \tag{25}$$

For any sensing matrix $\mathbf{A}$, output vector $\mathbf{y}$ and an unknown signal $\mathbf{x}$ generated from $\tilde{\mathbf{x}}$ using the above process, we have

$$
\begin{aligned}
I(\mathbf{A}, \mathbf{y}; \mathbf{x} | \tilde{\mathbf{x}}) &= H(\mathbf{x} | \tilde{\mathbf{x}}) - H(\mathbf{x} | \mathbf{A}, \mathbf{y}, \tilde{\mathbf{x}}) \\
&\geq H(\mathbf{x} | \tilde{\mathbf{x}}) - H(\mathbf{x} | \mathbf{A}, \mathbf{y}) \\
&\overset{(a)}{\geq} H(\mathbf{x} | \tilde{\mathbf{x}}) - h_2(\delta) - \delta \log \binom{n}{k} \\
&= \sum_{\tilde{\mathbf{x}}} \mathbb{P}\left(\tilde{\mathbf{x}} = \bar{\mathbf{x}}\right) H(\mathbf{x} | \tilde{\mathbf{x}} = \bar{\mathbf{x}}) - h_2(\delta) - \delta \log \binom{n}{k} \\
&\overset{(b)}{\geq} \sum_{\tilde{\mathbf{x}}} \mathbb{P}\left(\tilde{\mathbf{x}} = \bar{\mathbf{x}}\right) \log \binom{k}{l}\binom{n-k}{l} - h_2(\delta) - \delta k \log n \\
&\overset{(c)}{\geq} k h_2\left(\frac{l}{k}\right) + (n-k) h_2\left(\frac{l}{n-k}\right) - \log(k+1) \\
&\quad - \log(n-k+1) - h_2(\delta) - \delta k \log n \\
&\overset{(d)}{\geq} nN(l) - 2\log n - h_2(\delta) - \delta k \log n
\end{aligned} \tag{26}
$$

where $(a)$ follows from (Cover & Thomas, 2006, Theorem 2.10.1), $(b)$ follows from $\binom{n}{k} \leq n^k$, $(c)$ follows from $\binom{n}{k} \geq \frac{1}{n+1} 2^{n h_2(k/n)}$ ((Cover & Thomas, 2006, Theorem 11.1.3)) where $h_2$ is the binary entropy function and $(d)$ follows by defining $N(l) = \frac{k}{n} h_2\left(\frac{l}{k}\right) + (1 - \frac{k}{n}) h_2\left(\frac{l}{n-k}\right)$.

Next, we will compute an upper bound on $I(\mathbf{A}, \mathbf{y}; \mathbf{x} | \tilde{\mathbf{x}})$.

$$
\begin{aligned}
I(\mathbf{A}, \mathbf{y}; \mathbf{x} | \tilde{\mathbf{x}}) &= I(\mathbf{A}; \mathbf{x} | \tilde{\mathbf{x}}) + I(\mathbf{y}; \mathbf{x} | \mathbf{A}, \tilde{\mathbf{x}}) \\
&\overset{(a)}{=} 0 + I(\mathbf{y}; \mathbf{x} | \mathbf{A}, \tilde{\mathbf{x}}) \\
&\overset{(b)}{=} \sum_{i=1}^{m} I(y_i; \mathbf{x} | \mathbf{A}, y_{j \in [1:i-1]}, \tilde{\mathbf{x}})
\end{aligned}
$$

where $(a)$ follows because $\mathbf{A}$ is independent of both $\mathbf{x}$ and $\tilde{\mathbf{x}}$. In particular, $\mathbf{A}$ is conditionally independent of $\mathbf{x}$ conditioned on $\tilde{\mathbf{x}}$. Here, $(b)$ follows from chain rule for mutual information where $y_{j \in [1:i-1]}$ denotes $(y_1, \ldots, y_{i-1})$.

Suppose $h(\cdot)$ denotes the differential entropy of a continuous random variable. For any $i \in [1:m]$,

$$
\begin{aligned}
&I(y_i; \mathbf{x} | \mathbf{A}, y_{j \in [1:i-1]}, \tilde{\mathbf{x}}) \\
&= h(y_i | \mathbf{A}, y_{j \in [1:i-1]}, \tilde{\mathbf{x}}) - h(y_i | \mathbf{x}, \mathbf{A}, y_{j \in [1:i-1]}, \tilde{\mathbf{x}}) \\
&\overset{(a)}{\leq} h(y_i | \mathbf{A}_{i, j \in \mathcal{S}_{\tilde{\mathbf{x}}}}, \tilde{\mathbf{x}}) - h(\mathbf{A}_i^T \mathbf{x} + z_i | \mathbf{A}, \mathbf{x}, Y^{i-1}, \tilde{\mathbf{x}}) \\
&= h(y_i | \mathbf{A}_{i, j \in \mathcal{S}_{\tilde{\mathbf{x}}}}, \tilde{\mathbf{x}}) - h(z_i) \\
&= h(y_i | \mathbf{A}_{i, j \in \mathcal{S}_{\tilde{\mathbf{x}}}}, \tilde{\mathbf{x}}) - \frac{1}{2} \log\left(2\pi e \sigma^2\right)
\end{aligned}
$$

where in $(a)$, we use $\mathbf{A}_{i, j \in \mathcal{S}_{\tilde{\mathbf{x}}}}$ to denote the set of elements $A_{i,j}$ for $j \in \mathcal{S}_{\tilde{\mathbf{x}}}$. Conditioned on $\tilde{\mathbf{x}} = \bar{\mathbf{x}}$ and $\mathbf{A}_{i, j \in \mathcal{S}_{\tilde{\mathbf{x}}}} = \mathbf{a}_{i, j \in \mathcal{S}_{\tilde{\mathbf{x}}}}$,

$$h(y_i | \tilde{\mathbf{x}} = \bar{\mathbf{x}}, \mathbf{A}_{i,j \in \mathcal{S}_{\tilde{\mathbf{x}}}} = \mathbf{a}_{i,j \in \mathcal{S}_{\tilde{\mathbf{x}}}})$$

$$\overset{(a)}{=} h\left(y_i - \mathbb{E}\left[y_i | \tilde{\mathbf{x}} = \bar{\mathbf{x}}, \mathbf{A}_{\tilde{\mathbf{x}}} = \mathbf{a}_{i,j \in \mathcal{S}_{\tilde{\mathbf{x}}}}\right] \Big| \tilde{\mathbf{x}} = \bar{\mathbf{x}}, \mathbf{A}_{\tilde{\mathbf{x}}} = \mathbf{a}_{i,j \in \mathcal{S}_{\tilde{\mathbf{x}}}}\right)$$

$$\overset{(b)}{\leq} h(W_{\mathbf{a}_{i,j \in \mathcal{S}_{\tilde{\mathbf{x}}}}})$$

where $(a)$ follows by noting that differential entropy does not change by centering ((Cover & Thomas, 2006, Theorem 8.6.3)) and $(b)$ follows for $W_{i,\tilde{\mathbf{x}}} \sim \mathcal{N}\left(0, \sigma_w^2\right)$ where $\sigma_w^2 \leq \mathsf{Var}(y_i - \mathbb{E}\left[y_i | \tilde{\mathbf{x}} = \bar{\mathbf{x}}, \mathbf{A}_{i,j \in \mathcal{S}_{\tilde{\mathbf{x}}}} = \mathbf{a}_{i,j \in \mathcal{S}_{\tilde{\mathbf{x}}}}\right] \Big| \tilde{\mathbf{x}} = \bar{\mathbf{x}}, \mathbf{A}_{i,j \in \mathcal{S}_{\tilde{\mathbf{x}}}} = \mathbf{a}_{i,j \in \mathcal{S}_{\tilde{\mathbf{x}}}})$ from the fact that for the same variance a Gaussian random variable maximizes the differential entropy and it increasing with increasing variance ((Cover & Thomas, 2006, Theorem 8.6.5 and Example 8.1.2)).

Recall that each entry of $\mathbf{A}$ is chosen iid $\mathcal{N}(0, 1)$. In that case, $\mathsf{Var}\left(y_i - \mathbb{E}\left[y_i | \tilde{\mathbf{x}} = \bar{\mathbf{x}}, \mathbf{A}_{i,j \in \mathcal{S}_{\tilde{\mathbf{x}}}} = \mathbf{a}_{i,j \in \mathcal{S}_{\tilde{\mathbf{x}}}}\right]\right)$ conditioned on $\tilde{\mathbf{x}} = \bar{\mathbf{x}}$ and $\mathbf{A}_{\tilde{\mathbf{x}}} = \mathbf{a}_{i,j \in \mathcal{S}_{\tilde{\mathbf{x}}}}$ is given by $\mathbb{E}\left[(y_i)^2 | \tilde{\mathbf{x}} = \bar{\mathbf{x}}, \mathbf{A}_{i,j \in \mathcal{S}_{\tilde{\mathbf{x}}}} = \mathbf{a}_{i,j \in \mathcal{S}_{\tilde{\mathbf{x}}}}\right] - (\mathbb{E}\left[y_i | \tilde{\mathbf{x}} = \bar{\mathbf{x}}, \mathbf{A}_{i,j \in \mathcal{S}_{\tilde{\mathbf{x}}}} = \mathbf{a}_{i,j \in \mathcal{S}_{\tilde{\mathbf{x}}}}\right])^2$. We first analyse the first term.

$$\mathbb{E}\left[(y_i)^2 | \tilde{\mathbf{x}} = \bar{\mathbf{x}}, \mathbf{A}_{i,j \in \mathcal{S}_{\tilde{\mathbf{x}}}} = \mathbf{a}_{i,j \in \mathcal{S}_{\tilde{\mathbf{x}}}}\right]$$

$$= \mathbb{E}\left[\mathbb{E}\left[\left(\mathbf{A}_i^T \mathbf{x} + z_i\right)^2 | \tilde{\mathbf{x}} = \bar{\mathbf{x}}, \mathbf{A}_{i,j \in \mathcal{S}_{\tilde{\mathbf{x}}}} = \mathbf{a}_{i,j \in \mathcal{S}_{\tilde{\mathbf{x}}}}, \mathbf{x}\right]\right]$$

For any $\mathbf{x} = \hat{\mathbf{x}}$,

$$\mathbb{E}\left[\left(\mathbf{A}_i^T \mathbf{x} + z_i\right)^2 | \tilde{\mathbf{x}} = \bar{\mathbf{x}}, \mathbf{A}_{\tilde{\mathbf{x}}} = \mathbf{a}_{i,j \in \mathcal{S}_{\tilde{\mathbf{x}}}}, \mathbf{x} = \hat{\mathbf{x}}\right]$$

$$\overset{(a)}{=} l + \sigma^2 + (\mathbf{a}_{i, S_{\mathbf{x}} \cap S_{\tilde{\mathbf{x}}}})^2$$

where $(a)$ holds because conditioned on $\mathbf{A}_{\tilde{\mathbf{x}}} = \mathbf{a}_{i,j \in \mathcal{S}_{\tilde{\mathbf{x}}}}$ and $\mathbf{x} = \hat{\mathbf{x}}$, the random variable $\mathbf{A}_i^T \mathbf{x} + z_i = \mathbf{A}_{i, S_{\mathbf{x}} \setminus S_{\tilde{\mathbf{x}}}} + \mathbf{a}_{i, S_{\mathbf{x}} \cap S_{\tilde{\mathbf{x}}}} + z_i$ and $|S_{\mathbf{x}} \setminus S_{\tilde{\mathbf{x}}}| = l$.

Similarly, we can analyze the second term.

$$\mathbb{E}\left[y_i | \tilde{\mathbf{x}} = \bar{\mathbf{x}}, \mathbf{A}_{i,j \in \mathcal{S}_{\tilde{\mathbf{x}}}} = \mathbf{a}_{i,j \in \mathcal{S}_{\tilde{\mathbf{x}}}}\right]$$

$$= \mathbb{E}\left[\mathbf{A}_i^T \mathbf{x} + z_i | \tilde{\mathbf{x}} = \bar{\mathbf{x}}, \mathbf{A}_{i,j \in \mathcal{S}_{\tilde{\mathbf{x}}}} = \mathbf{a}_{i,j \in \mathcal{S}_{\tilde{\mathbf{x}}}}\right]$$

$$= \mathbb{E}\left[\mathbb{E}\left[\mathbf{A}_i^T \mathbf{x} + z_i | \tilde{\mathbf{x}} = \bar{\mathbf{x}}, \mathbf{A}_{i,j \in \mathcal{S}_{\tilde{\mathbf{x}}}} = \mathbf{a}_{i,j \in \mathcal{S}_{\tilde{\mathbf{x}}}}, \mathbf{x}\right]\right]$$

For any $\mathbf{x} = \hat{\mathbf{x}}$,

$$\mathbb{E}\left[\mathbf{A}_i^T \mathbf{x} + z_i | \tilde{\mathbf{x}} = \bar{\mathbf{x}}, \mathbf{A}_{i,j \in \mathcal{S}_{\tilde{\mathbf{x}}}} = \mathbf{a}_{i,j \in \mathcal{S}_{\tilde{\mathbf{x}}}}, \mathbf{x} = \hat{\mathbf{x}}\right]$$

$$= \mathbf{a}_{i, S_{\mathbf{x}} \cap S_{\tilde{\mathbf{x}}}}$$

and

$$\mathbb{E}\left[\mathbb{E}\left[\mathbf{Ax} + z_i | \tilde{\mathbf{x}} = \bar{\mathbf{x}}, \mathbf{A}_{i,j \in \mathcal{S}_{\tilde{\mathbf{x}}}} = \mathbf{a}_{i,j \in \mathcal{S}_{\tilde{\mathbf{x}}}}, \mathbf{x}\right]\right] = \mathbb{E}\left[\mathbb{E}\left[\mathbf{a}_{i, S_{\mathbf{x}} \cap S_{\tilde{\mathbf{x}}}} | \tilde{\mathbf{x}} = \bar{\mathbf{x}}, \mathbf{A}_{i,j \in \mathcal{S}_{\tilde{\mathbf{x}}}} = \mathbf{a}_{i,j \in \mathcal{S}_{\tilde{\mathbf{x}}}}, \mathbf{x}\right]\right]$$

$$\overset{(a)}{=} \frac{k - l}{k} \mathbf{a}_{i, S_{\tilde{\mathbf{x}}}}$$

where $(a)$ follows from (24).

$$\left(\mathbb{E}\left[y_i | \tilde{\mathbf{x}} = \bar{\mathbf{x}}, \mathbf{A}_{i,j \in \mathcal{S}_{\tilde{\mathbf{x}}}} = \mathbf{a}_{i,j \in \mathcal{S}_{\tilde{\mathbf{x}}}}\right]\right)^2$$

$$= \left(\frac{k - l}{k}\right)^2 \left(\sum_{j \in \mathcal{S}_{\tilde{\mathbf{x}}}} a_{i,j}^2 + 2 \sum_{\substack{j,l \in S_{\tilde{\mathbf{x}}} \\ j \neq l}} a_{i,j} a_{i,l}\right)$$

On the other hand,

$$\mathbb{E}\left[(y_i)^2 \,|\tilde{\mathbf{x}} = \bar{\mathbf{x}}, \mathbf{A}_{\tilde{\mathbf{x}}} = \mathbf{a}_{i,j \in \mathcal{S}_{\tilde{\mathbf{x}}}}\right]$$

$$= \mathbb{E}_{\mathbf{x}}\left[l + \sigma^2 + (\mathbf{a}_{i, S_{\mathbf{x}} \cap S_{\tilde{\mathbf{x}}}})^2\right]$$

$$\overset{(a)}{=} l + \sigma^2 + \frac{\binom{k-1}{k-l-1}}{\binom{k}{k-l}} \sum_{j \in S_{\tilde{\mathbf{x}}}} a_{i,j}^2 + 2\frac{\binom{k-2}{k-l-2}}{\binom{k}{k-l}} \sum_{\substack{j,l \in \mathcal{S}_{\tilde{\mathbf{x}}} \\ j \neq l}} a_{i,j} a_{i,l}$$

$$= l + \sigma^2 + \frac{k-l}{k} \sum_{j \in S_{\tilde{\mathbf{x}}}} a_{i,j}^2 + 2\left(\frac{k-l}{k}\right)\left(\frac{k-l-1}{k-1}\right) \sum_{\substack{j,l \in \mathcal{S}_{\tilde{\mathbf{x}}} \\ j \neq l}} a_{i,j} a_{i,l}$$

where $(a)$ follows from (24) and (25). Thus,

$$\text{Var}\left(y_i - \mathbb{E}\left[y_i | \tilde{\mathbf{x}} = \bar{\mathbf{x}}, \mathbf{A}_{i,j \in \mathcal{S}_{\tilde{\mathbf{x}}}} = \mathbf{a}_{i,j \in \mathcal{S}_{\tilde{\mathbf{x}}}}\right] \,\Big|\, \tilde{\mathbf{x}} = \bar{\mathbf{x}}, \mathbf{A}_{i,j \in \mathcal{S}_{\tilde{\mathbf{x}}}} = \mathbf{a}_{i,j \in \mathcal{S}_{\tilde{\mathbf{x}}}}\right)$$

$$= l + \sigma^2 + \frac{k-l}{k} \sum_{j \in S_{\tilde{\mathbf{x}}}} a_{i,j}^2 + 2\left(\frac{k-l}{k}\right)\left(\frac{k-l-1}{k-1}\right) \sum_{\substack{j,l \in \mathcal{S}_{\tilde{\mathbf{x}}} \\ j \neq l}} a_{i,j} a_{i,l} - \left(\frac{k-l}{k}\right)^2 \left(\sum_{j \in S_{\tilde{\mathbf{x}}}} a_{i,j}^2 + 2\sum_{\substack{j,l \in \mathcal{S}_{\tilde{\mathbf{x}}} \\ j \neq l}} a_{i,j} a_{i,l}\right)$$

$$= l + \sigma^2 + \left(\frac{k-l}{k}\right)\left(\frac{l}{k}\right) \sum_{j \in S_{\tilde{\mathbf{x}}}} a_{i,j}^2 - 2\frac{k-l}{k}\frac{l}{k(k-1)} \sum_{\substack{j,l \in \mathcal{S}_{\tilde{\mathbf{x}}} \\ j \neq l}} a_{i,j} a_{i,l}$$

Thus,

$$h(y_i | \mathbf{A}_{i,j \in \mathcal{S}_{\tilde{\mathbf{x}}}}, \tilde{\mathbf{x}} = \tilde{\mathbf{x}}) = \int p_{\mathbf{A}}(\mathbf{a}) h(y_i | \tilde{\mathbf{x}} = \bar{\mathbf{x}}, \mathbf{A}_{i,j \in \mathcal{S}_{\tilde{\mathbf{x}}}} = \mathbf{a}_{i,j \in \mathcal{S}_{\tilde{\mathbf{x}}}}) da$$

$$\leq \int p_{\mathbf{A}}(\mathbf{a}) \frac{1}{2} \log\left(2\pi e\left(l + \sigma^2 + \left(\frac{k-l}{k}\right)\left(\frac{l}{k}\right) \sum_{j \in S_{\tilde{\mathbf{x}}}} a_{i,j}^2 - 2\frac{k-l}{k}\frac{l}{k(k-1)} \sum_{\substack{j,l \in \mathcal{S}_{\tilde{\mathbf{x}}} \\ j \neq l}} a_{i,j} a_{i,l}\right)\right) d\mathbf{a}$$

$$\overset{(a)}{\leq} \frac{1}{2} \log\left(2\pi e\left(l + \sigma^2 + \left(\int p_{\mathbf{A}}(\mathbf{a}) \left(\frac{k-l}{k}\right)\left(\frac{l}{k}\right) \left(\sum_{j \in S_{\tilde{\mathbf{x}}}} a_{i,j}^2 - 2\frac{k-l}{k}\frac{l}{k(k-1)} \sum_{\substack{j,l \in \mathcal{S}_{\tilde{\mathbf{x}}} \\ j \neq l}} a_{i,j} a_{i,l}\right) d\mathbf{a}\right)\right)\right)$$

$$\overset{(b)}{=} \frac{1}{2} \log\left(2\pi e\left(l + \sigma^2 + \left(\frac{k-l}{k}\right)l\right)\right)$$

where $(a)$ follows from Jensen's inequality and $(b)$ follows by noting that $\mathbb{E}\left[A_{i,j}^2\right] = 1$ and $\mathbb{E}\left[A_{i,j} A_{i,l}\right] = 0$ for any $i$ and $j, l$, where $j \neq l$ and $\tilde{\mathbf{x}}$ is $k$-sparse.

Thus,

$$\sum_{i=1}^m I(y_i; \mathbf{x} | \mathbf{A}, y_{j \in [1:i-1]}, \tilde{\mathbf{x}}) \leq m\frac{1}{2} \log\left(2\pi e\left(l + \sigma^2 + \left(\frac{k-l}{k}\right)l\right)\right) - \frac{1}{2} \log\left(2\pi e\sigma^2\right)$$

$$= \frac{m}{2} \log\left(1 + \frac{l}{\sigma^2}\left(2 - \frac{l}{k}\right)\right)$$

Using this and (26), we conclude that

$$m \geq \frac{nN(l) - 2\log n - h_2(\delta) - \delta k \log n}{\frac{1}{2} \log\left(1 + \frac{l}{\sigma^2}\left(2 - \frac{l}{k}\right)\right)}.$$

$\square$

## B. Comparison with (Plan et al., 2017)

Algorithm 1 is similar to the two step estimation procedure outlined in (Plan et al., 2017) which was given to estimate the unknown signal within a two norm guarantee. Computing the vector $\mathbf{l} = (l_1, \ldots, l_n)$ is the same as the first step of the procedure in (Plan et al., 2017, Section 1.2) where a linear estimator is computed. The second step of our algorithm (sorting and keeping the top-$k$ indices) can be thought of as a projection on a feasible set (Plan et al., 2017, Section 1.3). However, this requires the estimation error to be small enough for the exact recovery of a binary vector.

The setup in (Plan et al., 2017) is for the recovery of an unknown signal with small two-norm error, whereas our problem of exact recovery of a sparse binary vector is more suited for recovery under infinity norm. This results in weak bounds ($m \approx O(k^2)$) when we specialize various results in (Plan et al., 2017) to our case. We first note that we require $\mathbb{E}\left\|\frac{\hat{x}}{\|\hat{x}\|_2} - \bar{x}\right\|_2 < \sqrt{\frac{2}{k}}$ for exact recovery. Otherwise, there exist two binary $k$-sparse vectors which have hamming distance at least two.

We first consider the 1-bit compressed sensing result in Section 3.5 (page 13). Setting the LHS to $\sqrt{\frac{2}{k}}$, we get

$$\sqrt{\frac{2}{k}} \leq C\sqrt{\frac{k \log (2n/k)}{m}}.$$

This implies that $m \approx C_1 k^2 \log (2n/k)$ for some constant $C_1$.

Next, we consider (Plan et al., 2017, Theorem 9.1). Note that for 1-bit compressed sensing $\eta^2 = 1$ and

$$\begin{aligned}
\mu &= \mathbb{E}\left[s_1 \langle a_1, \bar{x}\rangle\right] \\
&= \mathbb{E}\left[s_1 \langle a_1, x\rangle\right] \\
&\overset{(a)}{=} \frac{1}{\sqrt{k}}\sqrt{\frac{2}{\pi}} \times \frac{k}{\sqrt{(k+\sigma^2)}} \\
&= \sqrt{\frac{2}{\pi}} \times \frac{\sqrt{k}}{\sqrt{(k+\sigma^2)}}.
\end{aligned}$$

where $(a)$ follows from (17). Then,

$$\begin{aligned}
\|x - \mu\bar{x}\|_2 &= \left\|x - \sqrt{\frac{2}{\pi}} \times \frac{\sqrt{k}}{\sqrt{(k+\sigma^2)}}\frac{x}{\sqrt{k}}\right\|_2 \\
&= \left\|x - \sqrt{\frac{2}{\pi}} \times \frac{x}{\sqrt{(k+\sigma^2)}}\right\|_2
\end{aligned}$$

We require $\|x - \mu\bar{x}\|_2 < \frac{2}{\sqrt{\pi(k+\sigma^2)}}$ in order to exactly recover the unknown signal $x$.

We assume that $K$ is also a closed cone in $\mathbb{R}^n$. Then, by (Plan et al., 2017, Section 2.4), $w_t(K) = tw_1(K) \leq tC\sqrt{k \log (2n/k)}$ ((Plan et al., 2017, Section 2.4)). We choose $s = w_1(K)$. Substituting the bound for LHS and taking the limit $t \to 0$, we get

$$\frac{2}{\sqrt{\pi(k+\sigma^2)}} \leq \frac{8C\sqrt{k \log (2n/k)}}{\sqrt{m}}.$$

Thus, $m \approx 4C(k+\sigma^2)k \log (2n/k)$.

