# OpenReview forum: "Exact Recovery of Sparse Binary Vectors from Generalized Linear Measurements"
_ICML.cc/2025/Conference — ICML 2025 poster_

### Official Review · Reviewer_mHu2 · 2025-03-10

**Overall Recommendation:** 4

**Summary:**

The paper considers a new problem setting of recovering sparse binary vector from generalized linear measurements. The authors propose a simple algorithm based on Plan et al. (2017) and prove its performance guarantee, complemented by a nearly tight lower bound. This then implies tight resolution to noisy 1-bit compressed sensing and logistic regression. Interestingly, for SparseLinearReg, when $m = \Omega((k + \sigma^2) \log n)$, keeping the sign information only suffices. Also, for SparseLinearReg, the authors prove tighter upper and lower bounds based on MLE.

**Claims And Evidence:**

See Theoretical Claims.

**Essential References Not Discussed:**

None to the best of my knowledge.

**Experimental Designs Or Analyses:**

Not applicable.

**Methods And Evaluation Criteria:**

Not applicable.

**Other Comments Or Suggestions:**

Typos:
1. The notation $\bf{A}_i$ is used interchangeably between column vector and row vector throughout the paper. For instance, in Eqn. (2), $\bf{A}_i$ is the row-vector, but then shouldn't it be $\bf{A}_i \bf{x}$, not $\bf{A}_i^\top \bf{x}$? But then in Section 2.1, the same notation is overloaded with column vector. Also, in some parts, this is denoted as $A_i$ (e.g., line 360 left column).
2. Line 52 (first column): $[1 : m]$ is not defined
3. Line 91 (second column): Guassian => Gaussian
4. Line 128 (first column): Theorem 2.8 => Corollary 2.8
5. Line 639: what is the definition of $Q(\cdot)$?
6. Line 754: what is the definition of $h$? Actually, it seems that this is defined in line 978 as the differential entropy.... It should be defined sooner
7. Line 801: $\bf{x}' \Rightarrow \tilde{\bf{x}}$
8. Line 1085: Jenson => Jensen
9. In Theorem 2.10, $\max_l \Rightarrow \max_{l \in [1:k]}$

Suggestions:
1. I would suggest putting in a table comparing prior results and the results shown in this paper. Right now, all the comparisons to prior works are crammed into Section 1.1, which was a bit hard for me to parse at first, and I had to go back and forth after going through the theorems. Also, the table would help the readers organize the theorem results more clearly, as they are subtly different depending on the setting: specifically, whether $A_{i,j} \sim \mathcal{N}(0, 1)$ or not.
2. Some notations are left undefined, e.g., $\lVert \cdot \rVert_{\psi_2}$. The definitions should be included, at least in a footnote, for completeness.
3. Line 913: maybe put in a reference for the inequality (Theorem 11.1.3 of Cover & Thomas (2006))
4. Throughout the paper, I see things like [Example 2.5.8](Vershynin, 2018). If the authors meant to do \citet[Example 2.5.8]{vershynin2018highdim}, which should yield Vershynin (2018, Example 2.5.8), the authors should fix the bibtex errors accordingly.
5. Appendix B is too much not self-contained... The notations such as $s_1$, $K$, and $w_t$ aren't defined, as it seems that the reader must explicitly look them up in Plan et al. (2017)

**Other Strengths And Weaknesses:**

**Strengths:**
1. Clearly and well written
2. Simple yet effective resolution to the well-studied statistics problem with tight theoretical guarantees extending to GLM observations.
3. Although the proof flow itself is standard in information theory and statistics, there are several novel details that I appreciated: correlation vs. uncorrelation based on the support of $\bf{x}$ and appropriate use of independency arguments (virtually all theorems, to my understanding), coupling(?)-based tighter lower bound for SparseLinearReg (Theorem 2.10).

**Weakness:**
1. Given the algorithm's simplicity, it would have been better to include some toy experiments that showcase the tightness of the theoretical results presented.
2. Writing can be overall improved; see below.

**Questions For Authors:**

1. The proof of Theorem 2.1 doesn't seem to utilize the fact that the entry is Gaussian explicitly; rather, the only facts used were 1. $\lVert A_{i,j} \rVert_{\psi_2} \leq C'$ and 2. it satisfies the power constraint (Eqn. (2)). Then are there examples of subGaussian distribution (not Gaussian) that satisfy the power constraint? Then maybe Theorem 2.1 can be written in a more general sense. One immediate example may be binary sensing matrix, with $A_{i,j} \sim \mathrm{Ber}(p)$ for some $p \in (0, 1)$?
2. In Theorem 2.1, what is the meaning of $L$? This reminds me of the Polyak-Łojasiewicz condition, with a statistical twist(?). What do the authors think? Also, are there (subGaussian) GLMs that do not satisfy this $L$ condition? Lastly, has this $L$ condition been considered before in the GLM literature?
3. The statements of Corollary 2.2~2.4 should be either "... if $m = \Omega(...)$, or "there exists a $m_0 = O(...)$ such that when $m \geq m_0$, Algorithm 1 recovers the unknown signal".
4. The authors mention that the lower bound applies to the average error prob, while the upper bound is for the max prob. Then is there a chance that there may be a better algorithm (or better analysis) for deriving an upper bound for the average error prob, somewhat closing the gap between upper and lower bound ($k \log n$ vs. $k \log(n/k)$)? Or is the conjecture of Gamarnik & Zadik (2017a) basically saying that the gap (for now) is inevitable regardless of whether one looks at max error prob or average error prob?
5. By the way, in the abstract, the authors mention that there is no statistical-computational gap for 1bCSbinary and logistic regression. Why is this the case? To my understanding, the upper bounds hold for $m \gtrsim (k + \sigma^2) \log(nk)$ and the lower bound requires $m \gtrsim (k + \sigma^2) \log (n/k)$? ($1/\beta^2$ for logistic regression).
6. In the proof of Theorem 2.9 (Appendix A.3), can the authors elaborate on the last equality (equivalent formulation of log likelihood testing)?
7. Is there a combinatorial (or something similar) intuition behind $l = k \left( 1 - \frac{k}{n} \right)$?
8. It seems to me that the MLE-based approach is valid for GLM observations beyond SparseLinearReg, as we have a parametric assumption on the distribution, which makes $p(y|x)$ well-defined...? Then, am I correct in saying that the improved upper and lower bound arguments (Theorems 2.9, 2.10) cannot be extended trivially to GLM observations?

**Relation To Broader Scientific Literature:**

- To the best of my knowledge, tackles a new problem setting
- Nearly tight sample complexity bounds, which cannot be obtained by prior algorithms nor analyses
- Interesting proof techniques that may be of interest for statistics and information theory community

**Theoretical Claims:**

I checked all proofs and concur that they are generally correct. There are some minor issues and typos regarding absolute constants and tightness of the results, but the orderwise guarantees remain intact. Typos are deferred to the end. Here are some minor issues that I saw:

1. In line 338 (second column), it should be $y_i(A_{i,j} - A_{i,j'}) - E$, not $y_i(A_{i,j} - A_{i,j'}) - 1$.
2. In the proof of Theorem 2.5, when Fano's inequality is applied, why is there a "+1" in the $\log\left( \binom{n}{k} + 1 \right)$? To my understanding, the total number of possible k-sparse binary vectors is precisely $\binom{n}{k}$? Then one could bound $I$ as $I \geq (1 - \delta) k \log \frac{n}{k} - h_2(\delta)$, yielding a slightly tighter inequality.
3. In the last part of the proof of Theorem 2.5, when the authors invoke the result of Topsøe (2001), there is a $\log 2$ missing. Also, the range of $t$ should be $[-1/2, 1/2]$, not $[-1, 1]$.
4. In many parts of the other proofs (e.g., Proof of Theorem 2.10), the authors use the inequality $\binom{n}{k} \leq n^k$. It may be minor, but still I would prefer if the authors write the proofs and statements via $\binom{n}{k} \leq \left( \frac{en}{k} \right)^k$.

---

> ### Author Rebuttal · Authors · 2025-04-01
>
> Thank you very much for a careful review and all your interesting insights and questions.
> We completely agree with the suggestions, minor issues and typos mentioned. We will incorporate them in the next revision of the paper. In particular, we will fix all the minor issues and typos, incorporate the suggestions including adding a table of results and defining a separate notation for a row vector and column vector of ${A}$.
>
> Below we give responses to the questions asked.
>
> (Question 1) Your observation is correct. As you mentioned, there are other subgaussian distributions, for example, each entry of $A$ is chosen iid uniform on the set {0,1}  or {-1,1} that can be used as measurement matrices. The general proof technique will work for these distributions too. However, the theorem statement in its current form is true  only for a Gaussian matrix where each entry is chosen iid $\mathcal{N}(0,1)$. The proof uses the fact that each entry of the measurement matrix has zero mean and uses the distribution of the entries while applying Stein’s lemma between (11) and (12). As you noticed, the proof outline can easily be used for any other  distribution on the measurement matrix. We will mention it in the revised version of the paper.
>
> (Question 2) This is an interesting observation. While there might be some relation to an optimization paradigm, instead of this being a  condition on the model, it can  be thought of as a definition of $L$ which determines the rate of convergence. Note that we can also write  $L:= \min_{i\in [1:m]} \frac{\mathbb{E}(g'(A_i^Tx))}{||y_i|| }$  or can omit it altogether and write $\min_{i\in [1:m]} \frac{\mathbb{E}(g’(A_i^Tx))}{||y_i||_{\psi_2}}$ instead of $L$ in (7) (Theorem 2.1). Since $L$  could be vanishing with $k$ as is the case in this paper, as defined it will always exist.
> The numerator in $L$ can also be written as $\frac{\mathbb{E}(y_iA_i^Tx))}{k}$ (see the series of transformations between (11) and (12) on page 7). This quantity has previously been used in Plan,  Vershynin, Yudovina, 2017. See the definition of $\mu$ in proposition 1.1. This comes naturally while analyzing the convergence of linear estimator, since the estimator measures the contribution of each coordinate to the measurement outcomes.
>
> (Question 3) You are right. Thank you for noticing this.
>
>
> (Question 4) It turns out that for iid Gaussian design the avg error and max error criterions are equivalent.  Given any decoder $\phi$ with good average error performance, we can design $\phi'$ with good max error performance as follows:
>
> On input $(A,y)$, sample a uniform $n$ dimensional permutation matrix $R$; compute $\hat{x} = \phi(AR, y)$, and output $R\hat{x}$.
>
> Then, for any x, $\mathbb{P}\left(\phi'(A, Ax+z)\neq x\right) =\mathbb{P}\left(R\phi(AR, Ax+z)\neq x\right) =  \mathbb{P}\left(\phi(AR, ARR^{-1}x+z)\neq R^{-1}x\right) = \mathbb{P}\left(\phi(\tilde{A}, \tilde{A}R^{-1}x+z)\neq R^{-1}x\right) $ where $\tilde{A} = AR$. $AR$ is  iid Gaussian design because if we take a uniform permutation of columns of a gaussian matrix, it is still iid Gaussian. Since $\tilde{A}$ is i.i.d. Gaussian and $R^{-1}x$ is uniform $k$-sparse vector, this implies the max error of  $\phi'$ is same as the average error of $\phi$.
>
>
>
> Thus, the conjectured hardness by Gamarnik and Zadik is true for both maximum probability of error and average prob of error.
>
> We will mention it in the revised version.
>
> (Question 5) In the prior literature, $\log(n/k)$ and $\log{n}$ are not distinguished for the purpose of information-computation gap, which makes sense if $k = O(n^\alpha)$ for any $\alpha <1$. When $k = cn$ for some constant $c$, we can use an identity matrix to recover $x$ in $n$ measurements. Note that $k\log{n/k} = O(n)$ in this case. Therefore altogether $\log(n/k)$ and $\log{n}$ may not be differentiated for the purpose of this problem.
>
> (Question 6)  We believe that you are referring to  the last equality on page 15. We should have added another step here. This comes by substituting the value of probability densities as we show below. For any $r$ and set $\mathcal{V}$, the density  $p(y_r|A_{r,\mathcal{V}}) = \frac{1}{\sqrt{2\pi\sigma^2}}e^{-\frac{y_r-A_{r,\mathcal{V}}^2}{2\sigma^2}}$. Thus,   $$\mathbb{P}\left(\sum_{r}\log{\frac{p(y_r|A_{r,\mathcal{U}})}{p(y_r|A_{r,\mathcal{S}})}}>0\right)  = \mathbb{P}\left(\sum_{r}-\frac{(y_r-A_{r, \mathcal{U}})^2}{2\sigma^2} + \frac{(y_r-A_{r, \mathcal{S}})^2}{2\sigma^2} >0\right) $$ The last quantity here can be manipulated to get the last equality on page 15.
>
> (Question 7) You are right. In principle, the bounds of Theorem 2.9 and 2.10 can be extended to GLM observations. Though, it will be more difficult to analyse them as the probabilities will be more complex. We did not try to extend it for one bit compressed sensing and logistic regression because the given upper and lower bound already match up to constants. Recall that we can use an identity sensing matrix when $k$ is linear in $n$.

---

> > ### Comment · Reviewer_mHu2 · 2025-04-02
> >
> > I thank the authors for addressing all of my concerns and questions! I intend to keep my score and champion for its acceptance.
> >
> > Out of curiosity, could the authors respond to my Question #7 regarding the intuition behind $l = k \left( 1 - \frac{k}{n} \right)$? The authors' response to Question #7 seems to be for my Question #8. Thanks!
> >
> >
> > -------
> >
> > **After the authors' second rebuttal comment:**
> >
> > I'm satisfied with the authors' responses. Thank you, and congratulations on this nice work! I hope that the authors will incorporate all the comments and suggestions from me and other reviewers into the future revision, which will further strengthen the paper! I keep my score.

---

> > > ### Author Response · Authors · 2025-04-02
> > >
> > > We apologise for missing the answer to Question 7. Please find it below.
> > >
> > > This value of $l$ maximizes $N(l)$. To see this, note that $N(l)$ can be thought of as condition entropy $H(W|V)$ where $W$ and $V$ are both Bernoulli random variables, with $P(V=0) = k/n$ and $P(W=0|V=0) = 1-l/k$ and $P(W=0|V=1) = l/(n-k)$. The marginal on $W$ is given by $P(W=0)=k/n$.  Thus, $H(W|V) \leq H(W) = h_2(k/n)$. This upper bound is achieved by substituting  $l = k(1-k/n)$ in $N(l)$.
> > > Note that $2^{nN(l)}$ approximately gives the number of $k$-sparse vectors at Hamming distance $2l$ from a given vector. Therefore, this value of $l$, approximately,  gives the value of Hamming distance which maximizes the number of vectors at a particular hamming distance.
> > >
> > > We again thank you for your careful and positive review of the paper.

---

### Official Review · Reviewer_h1gw · 2025-03-11

**Overall Recommendation:** 4

**Summary:**

The paper studies the problem of recovering sparse binary vectors from noisy generalized linear measurements. For simplicity I am stating the special case problems SparseLinearReg and 1bCSbinary here: $x \in$ {$0,1$} is the unknown $k$-sparse vector that needs to be estimated. $A \in \mathbb{R}^{m \times n}$ is a sensing matrix (possibly random) with the power constraints $E[(A_i^\top x)^2] \leq k$. The $m$ linear observations using that matrix are the entries of the vector $y = Ax + z$ for SparseLinearReg or $y = sign(Ax + z)$ for 1bCSbinary. The goal is to use an appropriate $A$ and algorithm that takes $y$ and produces $x$ which is correct whp. The more general version of the problem uses generalized linear measurements using a link function in the standard way, which captures the above two problems as well as Logistic Regression as special cases. The paper uses $A$ having standard Gaussian entries and a very simple algorithm from Plan et al., 2017 that works by outputing the $k$ heaviest elements of the vector $(\langle y, A_i \rangle)_{i \in n}$ (which intuitively will correspond to the support of $x$ because $y$ is correlated with $A_i$ iff $x_i=1$). That way, the paper gives an upper bound for the sample complexity (achieved by the aforementioned computationally efficient algorithm) of the general problem, which yields corollaries for each of the three problems, SparseLinearReg, 1bCSbinary and LogisticRegression. Notably, if $m = \Theta((k+\sigma^2)(\log(k) + \log(n-k)))$ only the sign information in $y$ is enough in linear regression (i.e., 1bCSbinary is not harder than linear regression in that regime). The upper bounds also contradict the conjecture from Gamarnik and Zadik 2017 that no computationally efficient algorithm can exist for SparseLinearReg with the provided sample complexity. The paper provides an information theoretic lower bound based on Fano's inequality for the general problem which gives corollaries for each of the 3 problems. This gives tight characterization of the sample complexity for 1bCSbinary LogisticRegression. Finally, for exact recovery in SparseLinearReg, the authors complement the lower bound with an analysis of the MLE that gives almost a matching upper bound.

**Claims And Evidence:**

The main body and of the paper contains proofs for the claimed results.

**Essential References Not Discussed:**

I do not have additional suggestions.

**Experimental Designs Or Analyses:**

Not applicable.

**Methods And Evaluation Criteria:**

Not applicable.

**Other Comments Or Suggestions:**

.

**Other Strengths And Weaknesses:**

**Strenghts**
- Unified approach: the main result holds for the general problem that uses generalized linear measurements, and gives as corollaries the results for the three other problems.
- The algorithm is very simple to state and is computationally efficient
- It is interesting (and perhaps a bit surprising) that in the regime $m \approx (k + \sigma^2) \log n$ only the sign information suffices for sparse linear regression.
- The paper shows tight results for exact linear regression.

**Weaknesses**
- The gap between upper and lower bounds is hard to quantify, as the bound is expressed as some optimization problem.
- It remains open problem to find an efficient algorithm with better sample complexity than $(k + \sigma^2) \log n$ for linear regression.

Overall, I feel that the paper makes important contributions and would like to recommend acceptance.

**Questions For Authors:**

Related to the conjecture in Gamarnik and Zadik, 2017, do the authors think that a statistical-computational gap still exists but for some weaker threshold for the number of samples?

**Relation To Broader Scientific Literature:**

The paper discusses related work adequately.

**Theoretical Claims:**

I went through the main body of the paper and the claims seemed reasonable. That being said, I have not done a thorough check of all the proofs.

---

> ### Author Rebuttal · Authors · 2025-03-31
>
> Thank you very much for your careful and positive review of the paper.  Regarding your question, for the 1bCSbinary problem, there is no computational-statistical gap, since the information theoretic lower bound matches with the sample complexity of the linear estimator (up to constants). However the constant factors can be different, and also for other GLMs it may be the case that there is a computational statistical gap depending on the quantities $L$ (in Thm 2.1) and the mutual information $I(y_i, x|A)$ in Thm 2.5. Hope this answers your question.

---

### Official Review · Reviewer_g4Gp · 2025-03-13

**Overall Recommendation:** 4

**Summary:**

This paper addresses the problem of recovering a $k$-sparse binary vector from generalized linear measurements. Given observations $y = (y_1, \dots, y_m)$, which are related to a sparse vector $x$ through an inverse link function $g$ such that:

$$
\mathbb{E}[y_i | A_i] = g(A_i^T x), \quad \text{for each } i \in [m],
$$

the goal is to accurately reconstruct $x$. Since $x$ is binary, the problem reduces to support recovery.

It presents a simple “linear estimation followed by top‑k selection” algorithm – essentially a one‐shot version of an iterative greedy scheme – and provides tight sample complexity guarantees for several settings, including noisy one‑bit compressed sensing (1bCSbinary), sparse linear regression (SparseLinearReg), and logistic regression. In addition to the algorithmic upper bounds, the paper offers nearly matching information theoretic lower bounds. Notably, logistic regression and 1-bit compressed sensing emerge as special cases of this framework.

The authors build upon the simple linear estimation algorithm from Plan et al. (2017), focusing on its application to binary vectors. They establish both upper and lower bounds on the sample complexity required for successful recovery:

## Upper Bound: Sample Complexity of Algorithm 1 (Theorem 2.1)

If the generalized linear model (GLM) ensures that each $y_i$ is a subgaussian random variable with norm $\|y_i\|_{\psi_2}$, and for some $L$,

$$
\mathbb{E} \, g'(A_i^T x) \geq L \cdot \|y_i\|_{\psi_2} \quad \text{for all } i \in [m],
$$

then Algorithm 1 successfully recovers $x$ with high probability, provided the number of measurements satisfies:

$$
m \geq C \frac{\log(k) + \log(n-k)}{\min\{L, L^2\}},
$$

where $C$ is a constant.

## Lower Bound for GLMs (Theorem 2.5)

For any sensing matrix $A$, if $x$ is a uniformly chosen $k$-sparse vector, then any algorithm $\varphi$ that attempts to recover $x$ satisfies:

$$
P(\varphi(A, y) = x) \leq \delta
$$

only if the number of measurements meets the condition:

$$
m \geq k \log(n/k) \left[ 1 - h^2(\delta) + \delta k \log(n/k) \right],
$$

for some mutual information term $I$ satisfying $I \geq I(y_i; x | A)$ for all $i \in [m]$. Specifically, when $y$ takes binary values ($y \in \{-1, 1\}$), the expected squared inverse link function satisfies:

$$
\mathbb{E} [g(A_i^T x)^2] \geq I(y_i, x | A),
$$

where the expectation is taken over the randomness in $A$ and $x$.

## Update After Rebuttal:

I think the authors addressed my review sufficiently, and vote to accept the paper.

**Claims And Evidence:**

The claims made in the submission are supported by clear and convincing proofs.

**Essential References Not Discussed:**

I wonder if it might make sense to also cite work on lower bounds for sparse recovery (such as "Lower Bounds on Sparse Recovery" (Khanh Do Ba, Piotr Indyk, Eric Price, and David P. Woodruff)), since they also have these communication complexity style proofs.

**Experimental Designs Or Analyses:**

This paper does not have experiments.

**Methods And Evaluation Criteria:**

This is a theoretical paper, and this does not apply.

**Other Comments Or Suggestions:**

None

**Other Strengths And Weaknesses:**

### Strengths:

1. The paper has a unified treatment of multiple regression models.
2. The proposed algorithm -- linear estimator followed by top-k, is simple.
3. The insight into not having a stat-comp gap here is interesting.

### Weaknesses:

1. Many of the results make strong assumptions on the measurement matrices.
2. I feel the paper doesn't communicate the main novelty in analysis or algorithm clearly.

**Questions For Authors:**

N/A

**Relation To Broader Scientific Literature:**

The work builds on the foundational ideas from compressed sensing and sparse linear regression as developed by Candès et al. (2006), Donoho (2006), and Tibshirani (1996). The literature on one-bit compressed sensing—initiated by Boufounos and Baraniuk (2008) and later refined by Jacques et al. (2013)—has dealt with the challenges posed by extreme quantization (only sign information is available). This paper extends that line of work by incorporating Gaussian noise before quantization (the 1bCSbinary model), and it rigorously shows that the sample complexity remains optimal ($O((k+\sigma^2) \log n)$) even when only sign information is used.

**Theoretical Claims:**

I checked the first 9 pages and they seem fine to me.

---

> ### Author Rebuttal · Authors · 2025-04-01
>
> We thank you for your constructive and positive review.
>
> Thanks also for suggesting the additional reference. We will cite this paper as a relevant lower bound for sparse recovery. Since the input signal is binary in our case, the lower bounding techniques are somewhat different, as we can use information theoretic inequalities directly.
>
> Regarding the strong assumptions on the measurement matrices, we would like to mention that our lower bounds (sec 2.2 and Thm 2.5) work for any family of measurement matrices. Regarding the upper bound: we can make Thm 2.1 work for a more general class of matrices, at the expense of some clarity. Indeed, the only property of the Gaussian measurements that we use, conditioned on other assumptions of the theorem, is centered-ness, and Stein’s lemma. So the theorem holds for other matrices too, albeit will result in slightly complicated expressions (see some more details in the response to Reviewer mHu2).

---

### Decision · Program_Chairs · 2025-05-01

**Decision:**

Accept (poster)

**Comment:**

This paper studies the problem of exact recovery of sparse binary vectors from generalized linear measurements and presents a simple linear estimation algorithm with sample complexity bounds; there were also nearly matching info-theoretic lower bounds.

During the discussion, the authors authors addressed all comments raised to the reviewers' satisfaction. While no experiments were included, reviewers agreed this did not detract from the paper’s strong theoretical contributions. The strengths of the paper include its simplicity, clarity, and unification of several sparse recovery settings; this includes a nice insight that no stat-compute gap exists for 1bcs and logistic regression.

I encourage the authors to incorporate the reviewers’ suggestions to improve clarity; especially by fixing notation, defining constants carefully, and including a summary table of results. In all, this is a solid paper.